# Improving Sample Complexity Bounds for (Natural) Actor-Critic Algorithms

**Tengyu Xu, Zhe Wang, Yingbin Liang**

Department of ECE, The Ohio State University

{xu.3260,wang.10982,liang.889}@osu.edu

## Abstract

The actor-critic (AC) algorithm is a popular method to find an optimal policy in reinforcement learning. In the infinite horizon scenario, the finite-sample convergence rate for the AC and natural actor-critic (NAC) algorithms has been established recently, but under independent and identically distributed (i.i.d.) sampling and single-sample update at each iteration. In contrast, this paper characterizes the convergence rate and sample complexity of AC and NAC under Markovian sampling, with mini-batch data for each iteration, and with actor having general policy class approximation. We show that the overall sample complexity for a mini-batch AC to attain an $\epsilon$-accurate stationary point improves the best known sample complexity of AC by an order of $\mathcal{O}(\epsilon^{-1}\log(1/\epsilon))$, and the overall sample complexity for a mini-batch NAC to attain an $\epsilon$-accurate globally optimal point improves the existing sample complexity of NAC by an order of $\mathcal{O}(\epsilon^{-2}/\log(1/\epsilon))$. Moreover, the sample complexity of AC and NAC characterized in this work outperforms that of policy gradient (PG) and natural policy gradient (NPG) by a factor of $\mathcal{O}((1-\gamma)^{-3})$ and $\mathcal{O}((1-\gamma)^{-4}\epsilon^{-2}/\log(1/\epsilon))$, respectively. This is the first theoretical study establishing that AC and NAC attain orderwise performance improvement over PG and NPG under infinite horizon due to the incorporation of critic.

## 1 Introduction

The goal of reinforcement learning (RL) [39] is to maximize the expected total reward by taking actions according to a policy in a stochastic environment, which is modelled as a Markov decision process (MDP) [4]. To obtain an optimal policy, one popular method is the direct maximization of the expected total reward via gradient ascent, which is referred to as the policy gradient (PG) method [40, 47]. In practice, PG methods often suffer from large variance and high sampling cost caused by Monte Carlo rollouts to acquire the value function for estimating the policy gradient, which substantially slow down the convergence. To address such an issue, the actor-critic (AC) type of algorithms have been proposed [22, 23], in which *critic tracks the value function and actor updates the policy using the return of critic*. The usage of critic effectively reduces the variance of the policy update and the sampling cost, and significantly speeds up the convergence.

The first AC algorithm was proposed by [23], in which actor's updates adopt the simple stochastic policy gradient ascent step. This algorithm was later extended to the natural actor-critic (NAC) algorithm in [33, 9], in which actor's updates adopt the natural policy gradient (NPG) algorithm [18]. The asymptotic convergence of AC and NAC algorithms under both independent and identically distributed (i.i.d.) sampling and Markovian sampling have been established in [18, 21, 7, 9, 8]. The non-asymptotic convergence rate (i.e., the finite-sample analysis) of AC and NAC has recently been studied. More specifically, [54] studied the sample complexity of AC with linear function approximation in the linear quadratic regulator (LQR) problem. For general MDP with possibly

Table 1: Comparison of sample complexity of AC and NAC algorithms[1,2]

| Algorithm | Reference | Sampling | | Total complexity[3,4] |
| | | Actor | Critic | |
| --- | --- | --- | --- | --- |
| Actor-Critic | (Wang et al., 2019) [46] | i.i.d. | i.i.d. | $\mathcal{O}(\epsilon^{-4})$ |
| | (Kumar et al., 2019) [24] | i.i.d. | i.i.d. | $\mathcal{O}(\epsilon^{-4})$ |
| | (Qiu et al., 2019) [35] | i.i.d. | Markovian | $\mathcal{O}(\epsilon^{-3} \log^2(1/\epsilon))$ |
| | This paper | Markovian | Markovian | $\mathcal{O}(\epsilon^{-2} \log(1/\epsilon))$ |
| Natural Actor-Critic | (Wang et al., 2019) [46] | i.i.d. | i.i.d. | $\mathcal{O}(\epsilon^{-4})$ |
| | This paper | Markovian | Markovian | $\mathcal{O}(\epsilon^{-2} \log(1/\epsilon))$ |

[1] The table includes all previous studies on finite-sample analysis of AC and NAC under infinite-horizon MDP and policy function approximation, to our best knowledge.

[2] For comparison between our results of AC and NAC and the best known results of PG [49] and NPG [1], please refer to the discussion after Theorem 2 and Theorem 3.

[3] Total complexity of AC is measured to attain an $(\epsilon + \text{error})$-accurate stationary point $\bar{w}$, i.e., $\|\nabla_w J(\bar{w})\|_2^2 < \epsilon + \text{error}$. Total complexity of NAC is measured to attain an $(\epsilon + \text{error})$-accurate global optimum $\bar{w}$, i.e., $J(\pi^*) - J(\bar{w}) < \epsilon + \text{error}$.

[4] We do not include the dependence on $1 - \gamma$ into the complexity because most studies do not capture such dependence and it is difficult to make a fair comparison. Our results do capture such dependence as specified in our theorems.

infinity state space, [46] studied AC and NAC with both actor and critic utilize overparameterized neural networks as approximation functions, [24] studied AC with general nonlinear policy class and linear function approximation for critic, but with the requirement that the true value function is in the linear function class of critic. [35] studied a similar problem as [24] with weaker assumptions.

Although having progressed significantly, existing finite-sample analysis of AC and NAC have several limitations. They all assume that algorithms have access to the stationary distribution to generate i.i.d. samples, which can hardly be satisfied in practice. Moreover, existing studies focused on single-sample estimator for each update of actor and critic, which may not be overall sample-efficient.

- In this paper, we consider the *discounted* MDP with **infinite** horizon and possibly *infinite* state and action space, and with the policy taking a general *nonlinear* function approximation. We study the online AC and NAC algorithms, which has the entire execution based on a **single sample path** and each update based on a Markovian mini-batch of samples taken from such a sample path. We characterize the convergence rate for both AC and NAC, and show that mini-batch AC improves the best known sample complexity of AC [35] by a factor of $\mathcal{O}(\epsilon^{-1} \log(1/\epsilon))$ to attain an $\epsilon$-accurate stationary point, and mini-batch NAC improves the existing sample complexity [46] by a factor of $\mathcal{O}(\epsilon^{-2}/\log(1/\epsilon))$ to attain an $\epsilon$-accurate globally optimal point. Table 1 includes the detailed comparison among AC and NAC algorithms.

Second, the sample complexity of AC and NAC characterized in the existing studies is no better (in fact often worse) than that of PG and NPG under infinite horizon MDP. Specifically, the best known sample complexity $\mathcal{O}(\epsilon^{-3} \log^2(1/\epsilon))$ [35] of AC is worse than that of PG $\mathcal{O}(\epsilon^{-2})$ in [55, 49], and the best known complexity $\mathcal{O}(\epsilon^{-4})$ [46] of NAC is the same as that of NPG [1]. Clearly, these theoretical studies of AC and NAC did not capture their performance advantage over PG and NPG due to the incorporation of critic. Furthermore, the existing studies of AC and NAC with discounted reward did not capture the dependence of the sample complexity on $1 - \gamma$, and hence did not capture one important aspect of the comparison to PG and NPG.

- In this paper, for both AC and NAC, our characterization of the sample complexity is orderwisely better than the best known results for PG and NPG, respectively. Specifically, we show that AC improves the best known complexity $\mathcal{O}((1-\gamma)^{-5}\epsilon^{-2})$ of PG in [49] by a factor of $\mathcal{O}((1-\gamma)^{-3})$. We further show that NAC improves significantly upon the complexity $\mathcal{O}((1-\gamma)^{-8}\epsilon^{-4})$ of NPG in [1] by a factor of $\mathcal{O}((1-\gamma)^{-4}\epsilon^{-2}/\log(1/\epsilon))$. This is the *first* time that AC and NAC are shown to have better convergence rate than PG and NPG in theory.

We develop the following new techniques in our analysis. To obtain the convergence rate for critic, we develop a new technique to handle the bias error caused by *mini-batch* Markovian sampling in the linear stochastic approximation (SA) setting, which is different in nature from how existing studies handle single-sample bias [6]. Our result shows that Markovian mini-batch linear SA outperforms

single-sample linear SA in terms of the total sample complexity by a factor of $\log(1/\epsilon)$ [6, 38, 16]. For actor's update in AC, we develop a new technique to bound the bias error caused by mini-batch Markovian sampling in the **nonlinear SA** setting, which is different from the bias error of linear SA in critic's update. We show that the Markovian minibatch update allows a constant stepsize for actor's update, which yields a faster convergence rate and hence improves the total sample complexity by a factor of $\mathcal{O}(\epsilon^{-1})$ compared with previous study on AC [35]. For actor's update in NAC, we discover that the variance of actor's update is self-reduced under the Markovian mini-batch update, which yields an improved complexity by a factor of $\mathcal{O}(\epsilon^{-2})$ compared with previous study on NAC [46].

## 1.1 Related Work

We include here only the studies that are highly related to our work.

**AC and NAC.** The first AC algorithm was proposed by [23] and was later extended to NAC in [33] using NPG [18]. The asymptotic convergence of AC and NAC algorithms under both i.i.d. sampling and Markovian sampling have been established in [18, 21, 7, 9, 8]. The convergence rate (i.e., the finite-sample rate) of AC and NAC has been studied respectively in [46, 54, 24, 35] and in [46]. As aforementioned, all above convergence rate results are not better than that of PG and NPG. In contrast to the above studies of AC and NAC with single sample for each iteration, our study focuses on Markovian sampling and mini-batch data for each iteration, and establishes the improved sample complexity over the previous studies of AC and NAC. Two recent studies [52, 48] (concurrent to this paper) characterized the convergence rate of two time-scale AC in the Markovian setting. Our sample complexity also outperforms that of these two concurrent studies.

**Policy gradient.** The asymptotic convergence of PG in both the finite and infinite horizon scenarios has been established in [47, 3, 40, 18, 34, 42]. In some special RL problems such as LQR, under tabular policy, or with convex policy function approximation, PG has been shown to converge to the global optimum [15, 28, 45, 5]. General nonconcave/nonconvex function approximation has also been studied. For finite-horizon scenarios, [37, 31, 32, 50, 51] established the convergence rate (or sample complexity) of PG and variance reduced PG, and [12] studied the exploration efficiency of PG and established the regret bound. For infinite-horizon scenarios (which is the focus of this paper), [20] showed that PG converges to a neighborhood of a first-order stationary point and [55] modified the algorithm so that PG is guaranteed to converge to a second-order stationary point. The recent study [49] improved the sample complexity for PG in both studies [20, 55]. And [1] studied the convergence rate and sample complexity for NPG. This paper shows that AC and NAC have better convergence rate than the best known PG result in [49] and NPG result in [1]. As another line of research parallel to AC-type algorithms, more advanced PG algorithms TRPO/PPO have been studied in [36] for the tabular case and in [25] with the neural network function approximation.

**Linear SA and TD learning.** The convergence analysis of critic in AC and NAC in this paper is related to but different from the studies on TD learning, which we briefly summarize as follows. For TD learning under i.i.d. sampling (which can be modeled as linear SA with martingale noise), the asymptotic convergence has been well established in [11, 10], and the non-asymptotic convergence (i.e., finite-time analysis) has been provided in [13, 19, 43]. For TD learning under Markovian sampling (which can be modeled as linear SA with Markovian noise), the asymptotic convergence has been established in [44, 41], and the non-asymptotic analysis has been provided in [6, 53, 38, 16].

## 2 Problem Formulation and Preliminaries

In this section, we introduce the background of MDP, AC and NAC, and technical assumptions.

### 2.1 Markov Decision Process

A discounted Markov decision process (MDP) is defined by a tuple $(\mathcal{S}, \mathcal{A}, \mathsf{P}, r, \xi, \gamma)$, where $\mathcal{S}$ and $\mathcal{A}$ are the state and action spaces, $\mathsf{P}$ is the transition kernel, and $r$ is the reward function. Specifically, at step $t$, an agent takes an action $a_t \in \mathcal{A}$ at state $s_t \in \mathcal{S}$, transits into the next state $s_{t+1} \in \mathcal{S}$ according to the transition probability $\mathsf{P}(s_{t+1}|s_t, a_t)$ and receives a reward $r(s_t, a_t, s_{t+1})$. Moreover, $\xi$ denotes the distribution of the initial state $s_0 \in \mathcal{S}$ and $\gamma \in (0, 1)$ denotes the discount factor. A policy $\pi$ maps a state $s \in \mathcal{S}$ to the actions in $\mathcal{A}$ via a probability distribution $\pi(\cdot|s)$.

For a given policy $\pi$, we define the state value function as $V_\pi(s) = \mathbb{E}[\sum_{t=0}^\infty \gamma^t r(s_t, a_t, s_{t+1})|s_0 = s, \pi]$ and the state-action value function (i.e., the $Q$-function) as $Q_\pi(s,a) = \mathbb{E}[\sum_{t=0}^\infty \gamma^t r(s_t, a_t, s_{t+1})|s_0 = s, a_0 = a, \pi]$, where $a_t \sim \pi(\cdot|s_t)$ for all $t \geq 0$. We also define the advantage function of the policy $\pi$ as $A_\pi(s,a) = Q_\pi(s,a) - V_\pi(s)$. Moreover, the visitation measure induced by the police $\pi$ is defined as $\nu_\pi(s,a) = (1-\gamma)\sum_{t=0}^\infty \gamma^t \mathbb{P}(s_t = s, a_t = a)$. It has been shown in [21] that $\nu_\pi(s,a)$ is the stationary distribution of a Markov chain with the transition kernel $\widetilde{\mathsf{P}}(\cdot|s,a) = \gamma\mathsf{P}(\cdot|s,a) + (1-\gamma)\xi(\cdot)$ and the policy $\pi$ if the Markov chain is ergodic. For a given policy $\pi$, we define the expected total reward function as $J(\pi) = (1-\gamma)\mathbb{E}[\sum_{t=0}^\infty \gamma^t r(s_t, a_t, s_{t+1})] = \mathbb{E}_\xi[V_\pi(s)]$. The goal of reinforcement learning is to find an optimal policy $\pi^*$ that maximizes $J(\pi)$.

## 2.2  Policy Gradient Theorem

In order to find the optimal policy $\pi^*$ that maximizes $J(\pi)$, a popular approach is to parameterize the policy and then optimize over the set of parameters. We let the policy $\pi$ be parameterized by $w \in \mathcal{W} \subset \mathbb{R}^{d_1}$, where the parameter space $\mathcal{W}$ is Euclidean. Thus, the parameterized policy class is $\{\pi_w : w \in \mathcal{W}\}$. We allow general *nonlinear* parameterization of the policy $\pi$. Thus, the policy optimization problem is to solve the problem:

$$\max_{w \in \mathcal{W}} J(\pi_w) := J(w), \tag{1}$$

where we write $J(\pi_w) = J(w)$ for notational simplicity. In order to solve the problem eq. (1) by gradient-based approaches, the gradient $\nabla J(w)$ is derived by [40] as follows:

$$\nabla J(w) = \mathbb{E}_{\nu_{\pi_w}}\big[Q_{\pi_w}(s,a)\psi_w(s,a)\big] = \mathbb{E}_{\nu_{\pi_w}}\big[A_{\pi_w}(s,a)\psi_w(s,a)\big], \tag{2}$$

where $\psi_w(s,a) := \nabla_w \log \pi_w(a|s)$ denotes the score function. Ideally, policy gradient (PG) algorithms [47] update the parameter $w$ via gradient ascent: $w_{t+1} = w_t + \alpha\nabla_w J(w_t)$, where $\alpha > 0$ is the stepsize.

Alternatively, natural policy gradient (NPG) algorithms [18] apply natural gradient descent [2], which is invariant to the parametrization of policies. At each iteration, NPG ideally performs the update: $w_{t+1} = w_t + \alpha(F(w_t))^\dagger \nabla_w J(w_t)$, in which $F(w)$ is the Fisher information matrix given by $F(w) = \mathbb{E}_{\nu_{\pi_w}}[\psi_w(s,a)\psi_w(s,a)^\top]$. In practice, $F(w_t)$ is usually estimated via sampling [9].

In practice, both PG and NPG utilize Monte Carlo methods to estimate $Q_{\pi_w}(s,a)$ in eq. (2) to approximate the gradient $\nabla J(w_t)$. However, Monte Carlo rollout typically suffers from large variance and high sampling cost, which substantially degrades the convergence performance of PG and NPG. This motivates the design of *Actor-Critc (AC) and Natural Actor-Critic (NAC)* algorithms as we introduce in Section 2.3, which have significantly reduced variance and sampling cost.

## 2.3  Actor-Critic and Natural Actor-Critic Algorithms

We study the AC and NAC algorithms that adopt the design of Advantage Actor-Critic (A2C) proposed in [9, 30] (see Algorithm 1). Algorithm 1 performs **online** updates based on a **single sample path** in a nested fashion. Namely, the outer loop consists of actor's updates of the parameter $w$ to optimize the policy $\pi_w$, and each outer-loop update is followed by an entire inner loop of critic's $T_c$ updates of the parameter $\theta$ to estimate the value function $V_{\pi_w}(s)$, which further yields an estimate of the advantage function $A_{\pi_w}(s,a)$ to approximate the policy gradient in eq. (2).

**Critic's update:** Critic uses linear function approximation $V_\theta(s) = \phi(s)^\top\theta$ or $V_\theta = \Phi\theta$, and adopts TD learning to update the parameter $\theta$, where $\theta \in \mathbb{R}^{d_2}$, $\phi(\cdot): \mathcal{S} \to \mathbb{R}^{d_2}$ is a known feature mapping, and $\Phi$ is the correspondingly $|\mathcal{S}| \times d_2$ feature matrix. Critic updates the parameter $\theta$ as in Algorithm 2, which utilizes a mini-batch of samples $\{(s_{k,j}, a_{k,j}, s_{k,j+1})\}_{0 \leq j \leq M-1}$ sequentially drawn from the trajectory to perform the TD update (see line 8 of Algorithm 2).

**Actor's update:** Based on critic's estimation of the value function $V_\theta(s)$, actor approximates the advantage function $A_{\pi_w}(s,a)$ by the temporal difference error $\delta_\theta(s,a,s') = r(s,a,s') + \gamma V_\theta(s') - V_\theta(s)$. The policy gradient can then be estimated as $\nabla_w J(w) \approx \delta_\theta(s,a,s')\psi_w(s,a)$ based on eq. (2). In Algorithm 1, for AC, we adopt Markovian mini-batch sampling to estimate the policy gradient. For NAC, we first approximate the Fisher information matrix $F(w)$ via Markovian mini-batch sampling

---

**Algorithm 1** Actor-critic (AC) and natural actor-critic (NAC) online algorithms

---

1: **Input:** Policy class $\pi_w$, based function $\phi$, actor stepsize $\alpha$, critic stepsize $\beta$, regularization $\lambda$
2: **Initialize:** actor parameter $w_0$, initial state $s_0$
3: **for** $t = 0, \cdots, T - 1$ **do**
4:    $s_{\text{ini}} = s_{t-1,B}$ (when $t = 0$, $s_{\text{ini}} = s_0$)
5:    ***Critic update:*** $\theta_t, s_{t,0} = \text{Minibatch-TD}(s_{\text{ini}}, \pi_{w_t}, \phi, \beta, T_c, M)$
6:
7:    ***Online Markovian mini-batch sampling:***
8:    $F_t(w_t) = 0$
9:    **for** $i = 0, \cdots, B - 1$ **do**
10:       $a_{t,i} \sim \pi_{w_t}(\cdot|s_{t,i}), \quad s_{t,i+1} \sim \widetilde{\mathsf{P}}_{\pi_{w_t}}(\cdot|s_{t,i}, a_{t,i})$
11:       $\delta_{\theta_t}(s_{t,i}, a_{t,i}, s_{t,i+1}) = r(s_{t,i}, a_{t,i}, s_{t,i+1}) + \gamma\phi(s_{t,i+1})^\top \theta_t - \phi(s_{t,i})^\top \theta_t$
12:       $F_t(w_t) = F_t(w_t) + \frac{1}{B}\psi_{w_t}(s_{t,i}, a_{t,i})\psi_{w_t}^\top(s_{t,i}, a_{t,i})$ **(only for NAC update)**
13:    **end for**
14:
15:    ***Option I: Actor update in AC***
16:    $w_{t+1} = w_t + \alpha\frac{1}{B}\sum_{i=0}^{B-1}\delta_{\theta_t}(s_{t,i}, a_{t,i}, s_{t,i+1})\psi_{w_t}(s_{t,i}, a_{t,i})$
17:
18:    ***Option II: Actor update in NAC***
19:    $w_{t+1} = w_t + \alpha\left[F_t(w_t) + \lambda I\right]^{-1}\left[\frac{1}{B}\sum_{i=0}^{B-1}\delta_{\theta_t}(s_{t,i}, a_{t,i})\psi_{w_t}(s_{t,i}, a_{t,i}, s_{t,i+1})\right]$
20: **end for**
21: **Output:** $w_{\hat{T}}$ with $\hat{T}$ chosen uniformly from $\{1, \cdots, T\}$

---

---

**Algorithm 2** Minibatch-TD$(s_{\text{ini}}, \pi, \phi, \beta, T_c, M)$

---

1: **Initialize:** Critic parameter $\theta_0$
2: **for** $k = 0, \cdots, T_c - 1$ **do**
3:    $s_{k,0} = s_{k-1,M}$ ( when $k = 0$, $s_{k,0} = s_{\text{ini}}$)
4:    **for** $j = 0, \cdots, M - 1$ **do**
5:       $a_{k,j} \sim \pi(\cdot|s_{k,j}), \quad s_{k,j+1} \sim \mathsf{P}_\pi(\cdot|s_{k,j}, a_{k,j})$ (observe reward $r(s_{k,j}, a_{k,j}, s_{k,j+1})$)
6:       $\delta_{\theta_k}(s_{k,j}, a_{k,j}, s_{k,j+1}) = r(s_{k,j}, a_{k,j}, s_{k,j+1}) + \gamma\phi(s_{k,j+1})^\top \theta_k - \phi(s_{k,j})^\top \theta_k$
7:    **end for**
8:    **Critic update:** $\theta_{k+1} = \theta_k + \beta\frac{1}{M}\sum_{j=0}^{M-1}\delta_{\theta_k}(s_{k,j}, a_{k,j}, s_{k,j+1})\phi(s_{k,j})$
9: **end for**
10: **Output:** $\theta_{T_c}, s_{T_c-1,M}$

---

(see line 12 of Algorithm 1), where $\lambda I$ is the regularization term to prevent the matrix from being singular. We then update the policy parameter based on natural policy gradient.

In contrast to other nested-loop AC and NAC algorithms studied in [35, 24, 55, 1], which assume i.i.d. sampling, Algorithm 1 naturally takes a **single sample path** to perform the updates *without* requiring a restarted sample path. Specifically, critic inherits the sample path from the last iteration of actor to take the next Markovian sample (see lines 4 and 5 in Algorithm 1), and vice versa.

Note that our work is the *first* that applies the mini-batch technique to Markovian linear and nonlinear SA problems, which correspond respectively to critic and actor's iterations. We show in Section 3 that the mini-batch technique *orderwisely* improves the sample complexity of AC and NAC algorithms that apply single-sample update.

## 2.4 Technical Assumptions

We take the following standard assumptions throughout the paper.

**Assumption 1.** *For any $w, w' \in R^{d_1}$ and any state-action pair $(s, a) \in \mathcal{S} \times \mathcal{A}$, there exist positive constants $L_\phi$, $C_\phi$, and $C_\pi$ such that the following hold: (1) $\|\psi_w(s, a) - \psi_{w'}(s, a)\|_2 \leq L_\psi \|w - w'\|_2$; (2) $\|\psi_w(s, a)\|_2 \leq C_\psi$; (3) $\|\pi_w(\cdot|s) - \pi_{w'}(\cdot|s)\|_{TV} \leq C_\pi \|w - w'\|_2$, where $\|\cdot\|_{TV}$ denotes the total-variation norm.*

The first two items in Assumption 1 assume that the score function $\psi_w$ is smooth and bounded, which have also been adopted in previous studies [24, 55, 1, 21, 57]. The first two items can be satisfied by

many commonly used policy classes including some canonical policies such as Boltzman policy [22] and Gaussian policy [14]. The third item in Assumption 1 holds for any smooth policy with bounded action space or Gaussian policy. Lemma 1 in Appendix A provides such justifications.

**Assumption 2** (Ergodicity). *For any $w \in \mathbb{R}^{d_1}$, consider the MDP with policy $\pi_w$ and transition kernel $\mathsf{P}(\cdot|s,a)$ or $\widetilde{\mathsf{P}}(\cdot|s,a) = \gamma\mathsf{P}(\cdot|s,a) + (1-\gamma)\eta(\cdot)$, where $\eta(\cdot)$ can either be $\xi(\cdot)$ or $\mathsf{P}(\cdot|\hat{s},\hat{a})$ for any given $(\hat{s},\hat{a}) \in \mathcal{S} \times \mathcal{A}$. Let $\chi_{\pi_w}$ be the stationary distribution of the MDP. There exist constants $\kappa > 0$ and $\rho \in (0,1)$ such that*

$$\sup_{s \in \mathcal{S}} \|\mathbb{P}(s_t|s_0 = s) - \chi_{\pi_w}\|_{TV} \le \kappa\rho^t, \quad \forall t \ge 0,$$

*where $\mathbb{P}(s_t|s_0 = s)$ is the distribution of $s_t$ conditioned on $s_0 = s$.*

Assumption 2 has also been adopted in [6, 53, 57], which holds for any time-homogeneous Markov chain with finite-state space or any uniformly ergodic Markov chain with general state space.

# 3 Main Results

In this section, we first analyze the convergence of critic's update as a mini-batch linear SA algorithm. Based on such an analysis, we further provide the convergence rate for our AC and NAC algorithms.

## 3.1 Convergence Analysis of Critic: Mini-batch TD

In this section, we analyze critic's update, which adopts the mini-batch TD described in Algorithm 2 and can be viewed more generally as a mini-batch linear SA algorithm.

As we show below that *mini-batch* linear SA orderwisely improves the finite-time performance of the *single*-sample linear SA studied previously in [6, 38] in the Markovian setting. In fact, the finite-time analysis of *mini-batch* linear SA is very different from that of *single*-sample linear SA in [6, 38]. This is because samples in the same mini-batch are correlated with each other, which introduces an extra bias error within each iteration in addition to the bias error across iterations. Existing techniques such as in [6, 38] provide only ways to handle the correlation across iterations, but not the bias error within each iteration caused by a mini-batch Markovian data. Here, we develop a new analysis to handle such a bias error.

Specifically, we show that such a bias error can be divided into two parts, in which the first part diminishes as the algorithm approaches to the fix point, and the second part is averaged out as the batch size $M$ increases. Hence, the bias error can be controlled by the mini-batch size, so that mini-batch linear SA can converge arbitrarily close to the fix point with a *constant* stepsize chosen *independently* from the accuracy requirement. This orderwisely improves the sample complexity over single-sample linear SA [6, 38].

To present the convergence result, for any policy $\pi$, we define the matrix $A_\pi := \mathbb{E}_{\mu_\pi}[(\gamma\phi(s') - \phi(s))\phi(s)]$ and the vector $b_\pi := \mathbb{E}_{\mu_\pi}[r(s,a,s')\phi(s)]$. The optimal solution of TD learning $\theta_\pi^* = -A^{-1}b$. We assume that the feature mapping $\phi(s)$ is bounded for all $s \in \mathcal{S}$ and the columns of the feature matrix $\Phi$ are linearly independent. In such a case, it has been verified in [6, 45] that $(\theta - \theta_\pi^*)^\top A_\pi(\theta - \theta_\pi^*) \le -\lambda_{A_\pi}\|\theta - \theta_\pi^*\|_2^2$ for all $\theta \in \mathbb{R}^{d_2}$, where $\lambda_{A_\pi}$ is a positive constant.

The following theorem characterizes the convergence rate and sample complexity for Markovian mini-batch TD. The theorem is presented with the order-level terms to simplify the expression. The precise statement is provided as Theorem 4 (that includes Theorem 1 as a special case) in Appendix C together with the proof, which is for the general mini-batch linear SA with Markovian update.

**Theorem 1.** *Suppose Assumption 2 hold. Consider Algorithm 2 of Markovian mini-batch TD. Let stepsize $\beta = \min\{\mathcal{O}(\lambda_{A_\pi}), \mathcal{O}(\lambda_{A_\pi}^{-1})\}$. Then we have*

$$\mathbb{E}[\|\theta_{T_c} - \theta_\pi^*\|_2^2] \le (1 - \mathcal{O}(\lambda_{A_\pi}\beta))^{T_c} + \mathcal{O}\left(\frac{\beta}{M}\right).$$

*Let $T_c = \Theta(\log(1/\epsilon))$ and $M = \Theta(\epsilon^{-1})$. The total sample complexity for Algorithm 2 to achieve an $\epsilon$-accurate optimal solution $\theta_{T_c}$, i.e., $\mathbb{E}[\|\theta_{T_c} - \theta_\pi^*\|_2^2] \le \epsilon$, is given by $MT_c = \mathcal{O}(\epsilon^{-1}\log(1/\epsilon))$.*

**Comparison with existing results for TD:** Theorem 1 indicates that our mini-batch TD outperforms the best known sample complexity $\mathcal{O}(\epsilon^{-1}\log^2(1/\epsilon))$ of TD or linear SA in [6, 38] in the Markovian setting by a factor of $\mathcal{O}(\log(1/\epsilon))$. The utilization of mini-batch is crucial for such improvement, due to which the variance error of correlated samples decreases as the mini-batch size increases, and hence does not cause order-level increase in the total sample complexity.

## 3.2 Convergence Analysis of AC

In order to analyze AC algorithm, we first provide a property for $J(w)$.

**Proposition 1.** *Suppose Assumptions 1 and 2 hold. For any $w, w' \in \mathbb{R}^d$, we have $\|\nabla_w J(w) - \nabla_w J(w')\|_2 \leq L_J \|w - w'\|_2$, for all $w, w' \in \mathbb{R}^d$, where $L_J = (r_{\max}/(1 - \gamma))(4C_\nu C_\psi + L_\psi)$ and $C_\nu = (1/2)C_\pi \left(1 + \lceil \log_\rho \kappa^{-1}\rceil + (1-\rho)^{-1}\right)$.*

Proposition 1 has been given as the Lipschitz assumption in the previous studies of policy gradient and AC [24, 46], whereas we provide a proof as a formal justification for it to hold.

Since the objective function $J(w)$ in eq. (1) is nonconcave in general, the convergence analysis of AC is with respect to the standard metric of $\mathbb{E}\|\nabla_w J(w)\|_2^2$. Proposition 1 is thus crucial for such analysis. To present the convergence result of the our AC algorithm, we define the approximation error introduced by critic as $\zeta_{\text{approx}}^{\text{critic}} = \max_{w \in \mathcal{W}} \mathbb{E}_{\nu_w}[|V_{\pi_w}(s) - V_{\theta_{\pi_w}^*}(s)|^2]$. Such an error term also appears in the previous studies of AC [35, 9], or becomes zero under the assumption that the true value function $V_{\pi_w}(\cdot)$ belongs to the linear function space for all $w \in \mathcal{W}$ [24, 48].

The following theorem characterizes the convergence rate and sample complexity for our AC algorithm. The theorem is presented with the order-level terms to simplify the expression. The precise statement is provided as Theorem 5 in Appendix E together with the proof.

**Theorem 2.** *Consider the AC algorithm in Algorithm 1. Suppose Assumptions 1 and 2 hold, and let the stepsize $\alpha = \frac{1}{4L_J}$. Then we have*

$$\mathbb{E}[\|\nabla_w J(w_{\hat{T}})\|_2^2] \leq \mathcal{O}\left(\frac{1}{(1-\gamma)^2 T}\right) + \mathcal{O}\left(\frac{1}{T}\right)\sum_{t=0}^{T-1}\mathbb{E}[\|\theta_t - \theta_{\pi_{w_t}}^*\|_2^2] + \mathcal{O}\left(\frac{1}{B}\right) + \mathcal{O}(\zeta_{\text{approx}}^{\text{critic}}),$$

*Furthermore, let $B \geq \Theta(\epsilon^{-1})$ and $T \geq \Theta((1-\gamma)^{-2}\epsilon^{-1})$. Suppose the same setting of Theorem 1 holds (with $M$ and $T_c$ defined therein) so that $\mathbb{E}[\|\theta_t - \theta_{\pi_{w_t}}^*\|_2^2] \leq \mathcal{O}(\epsilon)$ for all $0 \leq t \leq T - 1$. Then we have*

$$\mathbb{E}[\|\nabla_w J(w_{\hat{T}})\|_2^2] \leq \epsilon + \mathcal{O}(\zeta_{\text{approx}}^{\text{critic}}),$$

*with the total sample complexity given by $(B + MT_c)T = \mathcal{O}((1-\gamma)^{-2}\epsilon^{-2}\log(1/\epsilon))$.*

The proof of Theorem 2 develops a new technique to handle the bias error for actor's update (which is nonlinear SA) due to Markovian mini-batch sampling. This is different from the bias error for critic's update (which is linear SA) that we handle in Theorem 1 and can be of independent interest.

**Comparison with existing results for AC:** Theorem 2 not only generalizes the previous studies [46, 24, 35] of single-sample AC under i.i.d. sampling to Markovian sampling, but also outperforms the best known sample complexity $\mathcal{O}(\epsilon^{-3}\log^2(1/\epsilon))$ in [35] by a factor of $\mathcal{O}(\epsilon^{-1}\log(1/\epsilon))$. Note that [35] does not study the discounted reward setting, and hence its result does not have the dependence on $1 - \gamma$. To explain where the improvement comes from, the mini-batch update plays two important roles here: **(1)** Previous studies use a single sample for actor's each update, and hence requires a *diminishing* stepsize to guarantee the convergence, which yields the convergence rate of $\mathcal{O}(1/\sqrt{T})$ [46, 24, 35]. In contrast, the mini-batch sampling allows a *constant* stepsize, and yields a faster convergence rate of $\mathcal{O}(1/T)$ and better overall sample complexity. **(2)** The mini-batch sampling keeps the bias error in actor's iteration at the same level of dependence on the mini-batch size $M$ as the variance error. In this way, the Markovian sampling does not cause order-level increase in the overall sample complexity.

**Comparison with existing results for PG:** The best known sample complexity of infinite horizon PG is given in Section 3.4 of [49], which is $\mathcal{O}((1-\gamma)^{-5}\epsilon^{-2})$. Clearly, Theorem 2 for mini-batch AC significantly outperforms such a result for PG by a factor of $\mathcal{O}((1-\gamma)^{-3}/\log(1/\epsilon))$, indicating that AC can converge much faster than vanilla PG. The heavy dependence of PG's complexity on

$1 - \gamma$ is caused by the utilization of Monte Carlo rollout to estimate the $Q$-function, which increases the sampling cost substantially and introduces large variance errors.

Theorem 2 is the *first* theoretical result establishing that AC algorithm outperforms PG in infinite horizon. The finite-sample analysis of AC algorithms in the previous studies [46, 24, 35] have worse dependence on $\epsilon$ than PG. In contrast, Theorem 2 shows that mini-batch AC has the same dependence on $\epsilon$ as PG (up to a logarithmic factor), but much better dependence on $1 - \gamma$, which often dominates the performance in RL scenarios.

### 3.3  Convergence Analysis of NAC

Differently from AC, due to the parameter invariant property of the NPG update, NAC can attain the globally optimal solution in terms of the function value convergence. In order to present the convergence guarantee of NAC, we define the estimation error introduced in actor's update $\zeta_{\text{approx}}^{\text{actor}} = \max_{w \in \mathcal{W}} \min_{p \in \mathbb{R}^{d_2}} \mathbb{E}_{\nu_{\pi_w}} \left[ \psi_w(s,a)^\top p - A_{\pi_w}(s,a) \right]^2$, which represents the approximation error caused by the insufficient expressive power of the parametrized policy class $\pi_w$. It can be shown that $\zeta_{\text{approx}}^{\text{actor}}$ is zero or small when the express power of the policy class $\pi_w$ is large, e.g., the tabular policy [1] and overparameterized neural policy [46].

The following theorem characterizes the convergence rate and sample complexity for our NAC algorithm. The theorem is presented with the order-level terms to simplify the expression. The precise statement is provided as Theorem 6 in Appendix F together with the proof.

**Theorem 3.** *Consider the NAC algorithm in Algorithm 1. Suppose Assumptions 1 and 2 hold, and let the stepsize $\alpha = \frac{\lambda^2}{4L_J(C_\psi^2 + \lambda)}$. Then we have*

$$J(\pi^*) - \mathbb{E}[J(\pi_{w_{\hat{T}}})] \leq \mathcal{O}\left(\frac{1}{(1-\gamma)^2 T}\right) + \mathcal{O}\left(\frac{1}{T}\right) \sum_{t=0}^{T-1} \mathbb{E}[\|\theta_t - \theta_{\pi_{w_t}}^*\|^2] + \mathcal{O}\left(\frac{1}{(1-\gamma)^2 B}\right)$$
$$+ \mathcal{O}\left(\frac{\sqrt{\zeta_{\text{approx}}^{\text{actor}}}}{(1-\gamma)^{1.5}}\right) + \mathcal{O}\left(\frac{\sqrt{\zeta_{\text{approx}}^{\text{critic}}}}{1-\gamma}\right) + \mathcal{O}\left(\frac{\zeta_{\text{approx}}^{\text{critic}}}{\lambda}\right) + \mathcal{O}\left(\frac{\lambda}{1-\gamma}\right),$$

*where $\lambda$ is the regularizing coefficient for estimating the inverse of Fisher information matrix. Furthermore, let $B \geq \Theta((1-\gamma)^{-2}\epsilon^{-1})$, $T \geq \Theta((1-\gamma)^{-2}\epsilon^{-1})$ and $\lambda = \mathcal{O}(\sqrt{\zeta_{\text{approx}}^{\text{critic}}})$. Suppose the same setting of Theorem 1 holds (with $M$ and $T_c$ defined therein) so that $\mathbb{E}[\|\theta_t - \theta_{\pi_{w_t}}^*\|^2] \leq \mathcal{O}(\epsilon)$ for all $0 \leq t \leq T-1$. Then we have*

$$J(\pi^*) - \mathbb{E}[J(\pi_{w_{\hat{T}}})] \leq \epsilon + \mathcal{O}\left(\frac{\sqrt{\zeta_{\text{approx}}^{\text{actor}}}}{(1-\gamma)^{1.5}}\right) + \mathcal{O}\left(\frac{\sqrt{\zeta_{\text{approx}}^{\text{critic}}}}{1-\gamma}\right),$$

*with the total sample complexity given by $(B + MT_c)T = \mathcal{O}((1-\gamma)^{-4}\epsilon^{-2}\log(1/\epsilon))$.*

Theorem 6 generalizes the previous study of NAC in [46] with i.i.d. sampling to that under Markovian sampling, and furthermore improves its sample complexity as we discuss below.

**Comparison with existing results of NAC:** The sample complexity of NAC was recently characterized in [46] as $\mathcal{O}(\epsilon^{-4})$, and the dependence on $(1-\gamma)$ was not captured. Theorem 3 improves their sample complexity by a factor of $\mathcal{O}(\epsilon^{-2}/\log(1/\epsilon))$, for which mini-batch sampling in both actor and critic's updates are crucial. Specifically, mini-batch sampling guarantees that even under a constant stepsize, the variance term in actor's update diminishes as both $\|\nabla_w J(w_t)\|_2$ and $\|\theta_t - \theta_{\pi_{w_t}}^*\|_2^2$ diminish, so that the global convergence follows. Thus, a constant stepsize yields a faster convergence rate of $\mathcal{O}(1/T)$ than $\mathcal{O}(1/\sqrt{T})$ of NAC in [46] due to diminishing stepsize.

**Comparison with existing results of NPG:** The sample complexity of NPG was recently characterized in [1, Corollary 6.10] as $\mathcal{O}((1-\gamma)^{-8}\epsilon^{-4})$. Clearly, Theorem 3 achieves better dependence on both $(1-\gamma)$ and $\epsilon$ than NPG by a factor of $\mathcal{O}((1-\gamma)^{-4}\epsilon^{-2}/\log(1/\epsilon))$. The novelty of our analysis is two folds. **(1)** Our analysis captures the benefit of the use of critic in NAC to estimate the value function rather than Monte Carlo rollout in NPG by a factor of $\mathcal{O}((1-\gamma)^{-4})$ saving in sample complexity, whereas the previous studies of NAC [46] does not capture the convergence dependence on $1 - \gamma$. **(2)** Our analysis exploits the self-reduction property of the variance error, so that a *constant* stepsize can be used to achieve a better complexity dependence on $\epsilon$.

## 4 Conclusion

In this paper, we provide the finite-sample analysis for mini-batch AC and NAC under Markovian sampling. This paper is the first work that applies the Markovian mini-batch technique to the AC and NAC algorithms and characterizes the performance improvement over the previous studies of these algorithms. Furthermore, this paper is also the first work that theoretically establishes the improvement of AC-type algorithms over PG-type algorithms by introduction of critic to reduce the variance and the sample complexity. For the future work, it is interesting to study the non-asymptotic convergence of AC-type algorithms in various settings such as multi-agent and distributed scenarios, with partial observations, under safety constraints, etc.

## Broader Impact

Policy optimization algorithms lie at the core of reinforcement learning, which has accomplished significant success in advancing technologies such as robotics, self-driving, online advertisement, etc. Among vast policy optimization algorithms, the actor-critic (AC) type of algorithms are broadly used and have achieved superior empirical performance. The focus of this paper is on exploring more sample-efficient AC-type algorithms and theoretically characterizing the advantage of the proposed schemes. We anticipate that these new innovations can be applied to other RL algorithms for performance improvement, such as Greedy-Q, nonlinear GTD [26], and off-policy AC algorithms [27, 56]. Ultimately, we hope that our sample saving ideas and techniques can be transferred into the real-world reinforcement learning technologies.

## Acknowledgement

The work was supported in part by the U.S. National Science Foundation under the grants CCF-1761506, CCF-1909291, CCF-1801855, and CCF-1900145.

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
