[Supplementary Material]

# Supplementary Materials

## A    Experiment

As suggested by one reviewer, we conduct the following experiment over Cartpole in OpenAI gym to show that the actor-critic algorithm with mini-batch updates can significantly outperform that with single-sample updates. We adopt neural softmax policy with two hidden layers of the size (128, 128). We apply the natural actor-critic (NAC) algorithm for updating the policy model, respectively with a single episode and with a mini-batch of episodes with the batchsize $B = 5$. The learning curves are shown in Figure 1. It can be seen that NAC with mini-batch episodes for each update has considerably faster convergence speed and more stable convergence performance than NAC with single episode for each update.

Figure 1: *The average performance for NAC over 10 seeds. The red and blue lines correspond to NAC with each update sampling 5 and 1 episodes, respectively.*

## B    Justification of Item 3 in Assumption 1

The following lemma justifies item 3 in Assumption 1. We denote the density function of the policy $\pi_w(\cdot|s)$ as $\frac{\pi_w(da|s)}{da}$ (if the action space $\mathcal{A}$ is discrete, then $\frac{\pi_w(da|s)}{da} = \pi_w(a|s)$).

**Lemma 1.** *Consider a policy $\pi_w$ parametrized by $w$. Consider the following two cases:*

1. *Density function of the policy is smooth, i.e. $\frac{\pi_w(da|s)}{da}$ is $L_\pi$-Lipschitz $(0 < L_\pi < \infty)$, and the action set is bounded, i.e. $\int_{a \in \mathcal{A}} 1 da = C_\mathcal{A} < \infty$,*

2. *$\pi_w$ is the Gaussian policy, i.e., $\pi_w(s) = \mathcal{N}(f(w), \sigma^2)$, with $f(w)$ being $L_f$-Lipschitz $(0 < L_f < \infty)$.*

*For both cases, we have*

$$\|\pi_w(\cdot|s) - \pi_{w'}(\cdot|s)\|_{TV} \leq C_\pi \|w - w'\|_2,$$

*where $C_\pi = \frac{1}{2} \max\{L_\pi C_\mathcal{A}, \sqrt{2} L_f\}$.*

*Proof.* Without loss of generality, we only consider the case when $\mathcal{A}$ is continuous. For the first case, we have

$$\|\pi_w(\cdot|s) - \pi_{w'}(\cdot|s)\|_{TV} = \frac{1}{2} \int_a \left| \frac{\pi_w(da|s)}{da} - \frac{\pi_{w'}(da|s)}{da} \right| da \overset{(i)}{\leq} \frac{1}{2} \int_a L_\pi \|w - w'\|_2 da$$

$$\leq \frac{1}{2} L_\pi C_\mathcal{A} \|w - w'\|_2 \leq C_\pi \|w - w'\|_2,$$

where $(i)$ follows from Assumption 1. For the second case, we have

$$\|\pi_w(\cdot|s) - \pi_{w'}(\cdot|s)\|_{TV} \leq \sqrt{\frac{1}{2}D_{KL}(\pi_w(\cdot|s), \pi_{w'}(\cdot|s))} = \sqrt{\frac{1}{2}(f(w) - f(w'))^2}$$

$$= \sqrt{\frac{1}{2}L_f^2 \|w - w'\|_2^2} = \frac{\sqrt{2}}{2}L_f \|w - w'\|_2 \leq C_\pi \|w - w'\|_2.$$

$\square$

## C  Proof of Proposition 1

By definition, we have

$$\nabla J(w) - \nabla J(w') = \int_{(s,a)} Q_{\pi_w}(s,a)\phi_w(s,a)\nu_{\pi_w}(ds, da) - \int_{(s,a)} Q_{\pi_{w'}}(s,a)\phi_{w'}(s,a)\nu_{\pi_{w'}}(ds, da)$$

$$= \int_{(s,a)} Q_{\pi_w}(s,a)\phi_w(s,a)\nu_{\pi_w}(ds, da) - \int_{(s,a)} Q_{\pi_w}(s,a)\phi_w(s,a)d\nu_{\pi_{w'}}(ds, da)$$

$$+ \int_{(s,a)} Q_{\pi_w}(s,a)\phi_w(s,a)d\nu_{\pi_{w'}}(ds, da) - \int_{(s,a)} Q_{\pi_{w'}}(s,a)\phi_{w'}(s,a)d\nu_{\pi_{w'}}(ds, da)$$

$$= \int_{(s,a)} Q_{\pi_w}(s,a)\phi_w(s,a)[\nu_{\pi_w}(ds, da) - \nu_{\pi_{w'}}(ds, da)]$$

$$+ \int_{(s,a)} [Q_{\pi_w}(s,a)\phi_w(s,a) - Q_{\pi_{w'}}(s,a)\phi_w(s,a)]\nu_{\pi_{w'}}(ds, da)$$

$$+ \int_{(s,a)} [Q_{\pi_{w'}}(s,a)\phi_w(s,a) - Q_{\pi_{w'}}(s,a)\phi_{w'}(s,a)]\nu_{\pi_{w'}}(ds, da).$$

Thus, we have

$$\|\nabla J(w) - \nabla J(w')\|_2 \leq \int_{(s,a)} \|Q_{\pi_w}(s,a)\phi_w(s,a)\|_2 \left|\nu_{\pi_w}(ds, da) - \nu_{\pi_{w'}}(ds, da)\right|$$

$$+ \int_{(s,a)} \left|Q_{\pi_w}(s,a) - Q_{\pi_{w'}}(s,a)\right| \|\phi_w(s,a)\|_2 \nu_{\pi_{w'}}(ds, da)$$

$$+ \int_{(s,a)} \left|Q_{\pi_{w'}}(s,a)\right| \|\phi_w(s,a) - \phi_{w'}(s,a)\|_2 \nu_{\pi_{w'}}(ds, da)$$

$$\leq \frac{r_{\max}C_\phi}{1 - \gamma} \int_{(s,a)} \left|\nu_{\pi_w}(ds, da) - \nu_{\pi_{w'}}(ds, da)\right|$$

$$+ C_\phi \int_{(s,a)} \left|Q_{\pi_w}(s,a) - Q_{\pi_{w'}}(s,a)\right| \nu_{\pi_{w'}}(ds, da)$$

$$+ \frac{r_{\max}}{1 - \gamma} \int_{(s,a)} \|\phi_w(s,a) - \phi_{w'}(s,a)\|_2 \nu_{\pi_{w'}}(ds, da)$$

$$\overset{(i)}{\leq} \frac{2r_{\max}C_\nu C_\phi}{1 - \gamma} \|w - w'\|_2 + \frac{2r_{\max}C_\nu C_\phi}{1 - \gamma} \|w - w'\|_2 + \frac{r_{\max}L_\phi}{1 - \gamma} \|w - w'\|_2$$

$$= L_J \|w - w'\|_2,$$

where $(i)$ follows from Lemma 3, Lemma 4 and Assumption 1.

## D  Proof of Theorem 1

In this section, we first provide the proof of a more general version (given as Theorem 4) of Theorem 1 for linear SA with Markovian mini-batch updates. We then show how Theorem 4 implies Theorem 1. Throughout the paper, for two matrices $M, N \in \mathbb{R}^{d \times d}$, we define $\langle M, N \rangle = \sum_{i=1}^{d} \sum_{j=1}^{d} M_{i,j} N_{i,j}$.

We consider the following linear stochastic approximation (SA) iteration with a constant stepsize:

$$\theta_{k+1} = \theta_k + \alpha\Big(\frac{1}{M}\sum_{i=kM}^{(k+1)M-1} A_{x_i}\theta_k + \frac{1}{M}\sum_{i=kM}^{(k+1)M-1} b_{x_i}\Big), \tag{3}$$

where $\{x_i\}_{i\geq 0}$ is a Markov chain with state space $\mathcal{X}$, and $A_{x_i} \in \mathbb{R}^{d\times d}$ and $b_{x_i} \in \mathbb{R}^d$ are random matrix and vector associated with $x_i$, respectively. We define $A = \mathbb{E}_\mu[A_x]$ and $b = \mathbb{E}_\mu[b_x]$, where $\mu$ is the stationary distribution of the associated Markov chain. Then the iteration eq. (3) corresponds to the following ODE:

$$\dot{\theta} = A\theta + b. \tag{4}$$

We consider the case when the matrix $A$ is non-singular, and we define $\theta^* = -A^{-1}b$ as the equilibrium point of the ODE in eq. (4). We make the following standard assumptions, which are also adopted by [6, 57, 53].

**Assumption 3.** *For all $x \in \mathcal{X}$, there exist constants such that the following hold*

1. *For all $x$, we have $\|A_x\|_F \leq C_A$ and $\|b_x\|_2 \leq C_b$,*

2. *There exist a positive constant $\lambda_A$ such that for any $\theta \in \mathbb{R}^d$, we have $\langle \theta - \theta^*, A(\theta - \theta^*)\rangle \leq -\frac{\lambda_A}{2}\|\theta - \theta^*\|_2^2$,*

3. *The MDP is irreducible and aperiodic, and there exist constants $\kappa > 0$ and $\rho \in (0,1)$ such that*

$$\sup_{x\in\mathcal{S}} \|\mathbb{P}(x_k \in \cdot | x_0) - \mu(\cdot)\|_{TV} \leq \kappa\rho^k, \quad \forall k \geq 0,$$

   *where $\mu(\cdot)$ is the stationary distribution of the MDP.*

It can be checked easily that if Assumption 3 holds, the equilibrium point $\theta^*$ has bounded $\ell_2$-norm, i.e., there exist a positive constant $R_\theta < \infty$ such that $\|\theta^*\|_2 \leq R_\theta$.

We first provide a lemma that is useful for the proof of the main theorem in this section.

**Lemma 2.** *Suppose Assumption 3 holds. Consider a Markov chain $\{x_i\}_{i\geq 0}$. Let $X_i$ be either $A_{x_i}$ or $b_{x_i}$, $C_x$ be either $C_A$ or $C_b$, respectively, and $\widetilde{X} = \mathbb{E}_\mu[X_x]$. For $t_0 \geq 0$ and $M > 0$, define $X(\mathcal{M}) = \frac{1}{M}\sum_{i=t_0}^{t_0+M-1} X(s_i)$. Then, we have*

$$\mathbb{E}\left[\left\|X(\mathcal{M}) - \widetilde{X}\right\|_2^2\right] \leq \frac{8C_x^2[1 + (\kappa - 1)\rho]}{(1 - \rho)M}.$$

*Proof.* We proceed as follows:

$$\mathbb{E}\left[\left\|X(\mathcal{M}) - \widetilde{X}\right\|_2^2 \Big| \mathcal{F}_{t_0}\right] \leq \mathbb{E}\left[\left\|X(\mathcal{M}) - \widetilde{X}\right\|_F^2 \Big| \mathcal{F}_{t_0}\right] = \mathbb{E}\left[\left\|\frac{1}{M}\sum_{i=t_0}^{t_0+M-1} X(s_i) - \widetilde{X}\right\|_F^2 \Big| \mathcal{F}_{t_0}\right]$$

$$\leq \frac{1}{M^2}\sum_{i=t_0}^{t_0+M-1}\sum_{j=t_0}^{t_0+M-1} \mathbb{E}\left[\langle X(s_i) - \widetilde{X}, X(s_j) - \widetilde{X}\rangle | \mathcal{F}_{t_0}\right]$$

$$\leq \frac{1}{M^2}\left[4MC_x^2 + \sum_{i\neq j}\mathbb{E}\left[\langle X(s_i) - \widetilde{X}, X(s_j) - \widetilde{X}\rangle | \mathcal{F}_{t_0}\right]\right]. \tag{5}$$

Consider the term $\mathbb{E}\left[\langle X(s_i) - \widetilde{X}, X(s_j) - \widetilde{X}\rangle | \mathcal{F}_{t_0}\right]$ with $i \neq j$. Without loss of generality, we consider the case when $i > j$:

$$\mathbb{E}\left[\langle X(s_i) - \widetilde{X}, X(s_j) - \widetilde{X}\rangle | \mathcal{F}_{t_0}\right]$$

$$= \mathbb{E}\left[\mathbb{E}[\langle X(s_i) - \widetilde{X}, X(s_j) - \widetilde{X}\rangle | s_j] | \mathcal{F}_{t_0}\right] = \mathbb{E}\left[\langle \mathbb{E}[X(s_i)|x_j] - \widetilde{X}, X(s_j) - \widetilde{X}\rangle | \mathcal{F}_{t_0}\right]$$

$$\leq \mathbb{E}\left[\left\|\mathbb{E}[X(s_i)|s_j] - \widetilde{X}\right\|_F \left\|X(s_j) - \widetilde{X}\right\|_F \Big| \mathcal{F}_{t_0}\right] \leq 2C_x \mathbb{E}\left[\left\|\mathbb{E}[X(s_i)|s_j] - \widetilde{X}\right\|_F \Big| \mathcal{F}_k\right]$$

$$\overset{(i)}{\leq} 4C_x^2 \kappa \rho^{j-i}, \tag{6}$$

where $(i)$ follows from Assumption 3 and the fact

$$\left\|\mathbb{E}[X(s_i)|s_j] - \widetilde{X}\right\|_F$$

$$= \left\|\int_{s_i} X(s_i)P(ds_i|s_j) - \int_{s_i} X(s_i)\nu(ds_i)\right\|_F \leq \int_{s_i} \|X(s_i)\|_F \left|P(ds_i|s_j) - \nu(ds_i)\right|$$

$$\leq C_x \int_{s_i} |P(ds_i|s_j) - \nu(ds_i)| \leq 2C_x \|P(\cdot|s_j) - \nu(\cdot)\|_{TV} \leq 2C_x \kappa \rho^{j-i}.$$

Substituting eq. (6) into eq. (5) yields

$$\mathbb{E}\left[\left\|X(\mathcal{M}) - \widetilde{X}\right\|_2^2 \Big| \mathcal{F}_{t_0}\right] \leq \frac{1}{M^2}\left[4MC_x^2 + 4C_x^2 \kappa \sum_{i\neq j} \rho^{|i-j|}\right] \leq \frac{8C_x^2[1+(\kappa-1)\rho]}{(1-\rho)M},$$

which completes the proof. $\qquad\square$

Now we proceed to prove the main theorem. For brevity, we use $\hat{A}_k$ and $\hat{b}_k$ to denote $\frac{1}{M}\sum_{i=kM}^{(k+1)M-1} A_{x_i}$ and $\frac{1}{M}\sum_{i=kM}^{(k+1)M-1} b_{x_i}$ respectively. We also define $g(\theta) = A\theta + b$ and $g_k(\theta) = \hat{A}_k\theta + \hat{b}_k$. We have the following theorem on the iteration of $\|\theta_K - \theta^*\|_2^2$.

**Theorem 4** (Generalized Version of Theorem 1). *Suppose Assumption 3 holds. Consider the iteration eq. (3). Let $\alpha \leq \min\{\frac{\lambda_A}{8C_A^2}, \frac{4}{\lambda_A}\}$ and $M \geq \left(\frac{2}{\lambda_A} + 2\alpha\right)\frac{192C_A^2[1+(\kappa-1)\rho]}{(1-\rho)\lambda_A}$. We have*

$$\mathbb{E}[\|\theta_K - \theta^*\|_2^2] \leq \left(1 - \frac{\lambda_A}{8}\alpha\right)^K \|\theta_0 - \theta^*\|_2^2 + \left(\frac{2}{\lambda_A} + 2\alpha\right)\frac{192(C_A^2 R_\theta^2 + C_b^2)[1+(\kappa-1)\rho]}{(1-\rho)\lambda_A M}.$$

*If we further let $K \geq \frac{8}{\lambda_A \alpha}\log\frac{2\|\theta_0 - \theta^*\|_2^2}{\epsilon}$ and $M \geq \left(\frac{2}{\lambda_A} + 2\alpha\right)\frac{384(C_A^2 R_\theta^2 + C_b^2)[1+(\kappa-1)\rho]}{(1-\rho)\lambda_A \epsilon}$, then we have $\mathbb{E}[\|\theta_K - \theta^*\|_2^2] \leq \epsilon$ with the total sample complexity given by $KM = \mathcal{O}\left(\frac{1}{\epsilon}\log\left(\frac{1}{\epsilon}\right)\right)$.*

**Proof of Theorem 4.** We first proceed as follows:

$$\begin{aligned}
\|\theta_{k+1} - \theta^*\|_2^2 &= \|\theta_k + \alpha g_k(\theta_k) - \theta^*\|_2^2 \\
&= \|\theta_k - \theta^*\|_2^2 + 2\alpha\langle\theta_k - \theta^*, g_k(\theta_k)\rangle + \alpha^2 \|g_k(\theta_k)\|_2^2 \\
&= \|\theta_k - \theta^*\|_2^2 + 2\alpha\langle\theta_k - \theta^*, g(\theta_k)\rangle + 2\alpha\langle\theta_k - \theta^*, g_k(\theta_k) - g(\theta_k)\rangle \\
&\quad + \alpha^2 \|g_k(\theta_k) - g(\theta_k) + g(\theta_k)\|_2^2 \\
&\overset{(i)}{\leq} \|\theta_k - \theta^*\|_2^2 - \lambda_A\alpha\|\theta_k - \theta^*\|_2^2 + \frac{\lambda_A}{2}\alpha\|\theta_k - \theta^*\|_2^2 + \frac{2}{\lambda_A}\alpha\|g_k(\theta_k) - g(\theta_k)\|_2^2 \\
&\quad + 2\alpha^2 \|g_k(\theta_k) - g(\theta_k)\|_2 + 2\alpha^2 \|g(\theta_k)\|_2^2 \\
&\overset{(ii)}{\leq} \left(1 - \frac{\lambda_A}{2}\alpha + 2C_A^2\alpha^2\right)\|\theta_k - \theta^*\|_2^2 + \left(\frac{2}{\lambda_A}\alpha + 2\alpha^2\right)\|g_k(\theta_k) - g(\theta_k)\|_2^2, \quad (7)
\end{aligned}$$

where $(i)$ follows from the facts that

$$\langle\theta_k - \theta^*, g(\theta_k)\rangle = \langle\theta_k - \theta^*, A(\theta_k - \theta^*)\rangle \leq -\frac{\lambda_A}{2}\|\theta_k - \theta^*\|_2^2,$$

$$\langle\theta_k - \theta^*, g_k(\theta_k) - g(\theta_k)\rangle \leq \frac{\lambda_A}{4}\|\theta_k - \theta^*\|_2^2 + \frac{1}{\lambda_A}\|g_k(\theta_k) - g(\theta_k)\|_2^2,$$

and

$$\|g_k(\theta_k) - g(\theta_k) + g(\theta_k)\|_2^2 \leq 2\|g_k(\theta_k) - g(\theta_k)\|_2^2 + 2\|g(\theta_k)\|_2^2,$$

and (ii) follows from the fact that $\|g(\theta_k)\|_2 = \|A(\theta_k - \theta^*)\|_2 \leq C_A \|\theta_k - \theta^*\|_2$. Let $\mathcal{F}_k$ be the filtration of the sample $\{x_i\}_{0 \leq i \leq kM-1}$. Taking expectation on both sides of eq. (7) conditioned on $\mathcal{F}_k$ yields

$$\mathbb{E}[\|\theta_{k+1} - \theta^*\|_2^2 \,|\mathcal{F}_k]$$
$$\leq \left(1 - \frac{\lambda_A}{2}\alpha + 2C_A^2\alpha^2\right) \|\theta_k - \theta^*\|_2^2 + \left(\frac{2}{\lambda_A}\alpha + 2\alpha^2\right)\mathbb{E}[\|g_k(\theta_k) - g(\theta_k)\|_2^2 \,|\mathcal{F}_k]. \quad (8)$$

Next we bound the term $\mathbb{E}[\|g_k(\theta_k) - g(\theta_k)\|_2^2 \,|\mathcal{F}_k]$ in eq. (8) as follows.

$$\mathbb{E}[\|g_k(\theta_k) - g(\theta_k)\|_2^2 \,|\mathcal{F}_k]$$
$$= \mathbb{E}\left[\left\|(\hat{A}_k - A)\theta_k + \hat{b}_k - b\right\|_2^2 \,\Big|\, \mathcal{F}_k\right]$$
$$= \mathbb{E}\left[\left\|(\hat{A}_k - A)(\theta_k - \theta^*) + (\hat{A}_k - A)\theta^* + \hat{b}_k - b\right\|_2^2 \,\Big|\, \mathcal{F}_k\right]$$
$$\leq 3\mathbb{E}\left[\left\|(\hat{A}_k - A)(\theta_k - \theta^*)\right\|_2^2 + \left\|(\hat{A}_k - A)\theta^*\right\|_2^2 + \left\|\hat{b}_k - b\right\|_2^2 \,\Big|\, \mathcal{F}_k\right]$$
$$\leq 3\mathbb{E}\left[\left\|\hat{A}_k - A\right\|_2^2 \,\Big|\, \mathcal{F}_k\right]\|\theta_k - \theta^*\|_2^2 + 3\mathbb{E}\left[\left\|\hat{A}_k - A\right\|_2^2 \,\Big|\, \mathcal{F}_k\right]\|\theta^*\|_2^2 + 3\mathbb{E}\left[\left\|\hat{b}_k - b\right\|_2^2 \,\Big|\, \mathcal{F}_k\right]. \quad (9)$$

Following from Lemma 2, we obtain

$$\mathbb{E}\left[\left\|\hat{A}_k - A\right\|_2^2 \,\Big|\, \mathcal{F}_k\right] \leq \frac{1}{M^2}\left[4MC_A^2 + 4C_A^2\kappa \sum_{i \neq j} \rho^{|i-j|}\right] \leq \frac{8C_A^2[1 + (\kappa - 1)\rho]}{(1 - \rho)M}, \quad (10)$$

and

$$\mathbb{E}\left[\left\|\hat{b}_k - b\right\|_2^2 \,\Big|\, \mathcal{F}_t\right] \leq \frac{8C_b^2[1 + (\kappa - 1)\rho]}{(1 - \rho)M}. \quad (11)$$

Substituting eq. (10) and eq. (11) into eq. (9) yields

$$\mathbb{E}[\|g_k(\theta_k) - g(\theta_k)\|_2^2 \,|\mathcal{F}_k] \leq \frac{24C_A^2[1 + (\kappa - 1)\rho]}{(1 - \rho)M} \|\theta_k - \theta^*\|_2^2 + \frac{24(C_A^2 R_\theta^2 + C_b^2)[1 + (\kappa - 1)\rho]}{(1 - \rho)M}. \quad (12)$$

Then, substituting eq. (12) into eq. (7) yields

$$\mathbb{E}[\|\theta_{k+1} - \theta^*\|_2^2 \,|\mathcal{F}_k] \leq \left(1 - \frac{\lambda_A}{2}\alpha + 2C_A^2\alpha^2 + \left(\frac{2}{\lambda_A}\alpha + 2\alpha^2\right)\frac{24C_A^2[1 + (\kappa - 1)\rho]}{(1 - \rho)M}\right) \|\theta_k - \theta^*\|_2^2$$
$$+ \left(\frac{2}{\lambda_A}\alpha + 2\alpha^2\right)\frac{24(C_A^2 R_\theta^2 + C_b^2)[1 + (\kappa - 1)\rho]}{(1 - \rho)M}.$$

Letting $\alpha \leq \frac{\lambda_A}{8C_A^2}$ and $M \geq \left(\frac{2}{\lambda_A} + 2\alpha\right)\frac{192C_A^2[1+(\kappa-1)\rho]}{(1-\rho)\lambda_A}$, and taking expectation over $\mathcal{F}_t$ on both sides of the above inequality yield

$$\mathbb{E}[\|\theta_{k+1} - \theta^*\|_2^2] \leq \left(1 - \frac{\lambda_A}{8}\alpha\right)\mathbb{E}[\|\theta_k - \theta^*\|_2^2] + \left(\frac{2}{\lambda_A}\alpha + 2\alpha^2\right)\frac{24(C_A^2 R_\theta^2 + C_b^2)[1 + (\kappa - 1)\rho]}{(1 - \rho)M}. \quad (13)$$

Applying eq. (13) recursively from $k = 0$ to $K - 1$ and letting $\alpha < \frac{8}{\lambda_A}$ yield

$$\mathbb{E}[\|\theta_K - \theta^*\|_2^2]$$
$$\leq \left(1 - \frac{\lambda_A}{8}\alpha\right)^K \|\theta_0 - \theta^*\|_2^2 + \left(\frac{2}{\lambda_A}\alpha + 2\alpha^2\right)\frac{24(C_A^2 R_\theta^2 + C_b^2)[1 + (\kappa - 1)\rho]}{(1 - \rho)M} \sum_{k=0}^{K-1} \left(1 - \frac{\lambda_A}{8}\alpha\right)^k$$

$$\leq \left(1 - \frac{\lambda_A}{8}\alpha\right)^K \|\theta_0 - \theta^*\|_2^2 + \left(\frac{2}{\lambda_A} + 2\alpha\right)\frac{192(C_A^2 R_\theta^2 + C_b^2)[1 + (\kappa - 1)\rho]}{(1 - \rho)\lambda_A M}$$

$$\leq e^{-\frac{\lambda_A}{8}\alpha K} \|\theta_0 - \theta^*\|_2^2 + \left(\frac{2}{\lambda_A} + 2\alpha\right)\frac{192(C_A^2 R_\theta^2 + C_b^2)[1 + (\kappa - 1)\rho]}{(1 - \rho)\lambda_A M}. \tag{14}$$

Letting $\alpha = \min\{\frac{\lambda_A}{8C_A^2}, \frac{4}{\lambda_A}\}$, $K \geq \frac{8}{\lambda_A \alpha}\log\frac{2\|\theta_0 - \theta^*\|_2^2}{\epsilon}$ and $M \geq \left(\frac{2}{\lambda_A} + 2\alpha\right)\frac{384(C_A^2 R_\theta^2 + C_b^2)[1 + (\kappa - 1)\rho]}{(1 - \rho)\lambda_A \epsilon}$, we have $\mathbb{E}[\|\theta_K - \theta^*\|_2^2] \leq \epsilon$. $\qquad\square$

Then, We show how to apply Theorem 4 to derive the sample complexity of Algorithm 2 given in Theorem 1.

**Proof of Theorem 1.** We define the parameters in Theorem 4 to be $A_{x_i} = \phi(s_{t,i})(\gamma\phi(s_{t,i+1}) - \phi(s_{t,i}))^\top$, $b_{x_i} = r(s_{t,i}, a_{t,i}, s_{t,i+1})\phi(s_{t,i})$ and $K = T_c$. Then the results of Theorem 1 follows. $\qquad\square$

# E   Supporting Lemmas for Theorem 2 and Theorem 3

In this subsection, we provide supporting lemmas, which are useful to the proof of Theorem 2.

**Lemma 3.** *Consider the initialization distribution $\eta(\cdot)$ and transition kernel $\mathsf{P}(\cdot|s,a)$. Let $\eta(\cdot) = \zeta(\cdot)$ or $\mathsf{P}(\cdot|\hat{s}, \hat{a})$ for any given $(\hat{s}, \hat{a}) \in \mathcal{S} \times \mathcal{A}$. Denote $\nu_{\pi_w, \eta}(\cdot, \cdot)$ as the state-action visitation distribution of MDP with policy $\pi_w$ and initialization distribution $\eta(\cdot)$. Suppose Assumption 2 holds. Then we have*

$$\left\|\nu_{\pi_w, \eta}(\cdot, \cdot) - \nu_{\pi_{w'}, \eta}(\cdot, \cdot)\right\|_{TV} \leq C_\nu \|w - w'\|_2$$

*for all $w, w' \in \mathbb{R}^d$, where $C_\nu = C_\pi \left(1 + \lceil\log_\rho \kappa^{-1}\rceil + \frac{1}{1-\rho}\right)$.*

*Proof.* The proof of this lemma is similar to the proof of Lemma 6 in [57] with the following difference. [57] considers the case with the finite action space, we extend their result to the case with possibly infinite action space. Define the transition kernel $\widetilde{\mathsf{P}}(\cdot|s,a) = \gamma\mathsf{P}(\cdot|s,a) + (1 - \gamma)I(\cdot)$. Denote $P_{\pi_w, I}(\cdot)$ as the state visitation distribution of the MDP with policy $\pi_w$ and initialization distribution $I(\cdot)$, and it satisfies that $\nu_{\pi_w, I}(s, a) = P_{\pi_w, I}(s)\pi_w(a|s)$. [21] showed that the stationary distribution of the MDP with transition kernel $\widetilde{\mathsf{P}}(\cdot|s,a)$ and policy $\pi_w$ is given by $P_{\pi_w, I}(\cdot)$. Following from Theorem 3.1 in [29], we obtain

$$\left\|P_{\pi_w, I}(\cdot) - P_{\pi_{w'}, I}(\cdot)\right\|_{TV} \leq \left(\lceil\log_\rho \kappa^{-1}\rceil + \frac{1}{1-\rho}\right)\|K_w - K_{w'}\|, \tag{15}$$

where $K_w$ and $K_{w'}$ are state to state transition kernel of MDP with policy $\pi_w$ and $\pi_{w'}$ respectively and $\|\cdot\|$ is the operator norm of a transition kernel: $\|P\| := \sup_{\|q\|_{TV}=1}\|qP\|_{TV}$. Note here we define the total variation norm of a distribution $q(s)$ as $\|q\|_{TV} = \int_s |q(ds)|$. Then we obtain

$$\|K_w - K_{w'}\| = \sup_{\|q\|_{TV}=1}\left\|\int_s q(ds)(K_w - K_{w'})(s, \cdot)\right\|_{TV}$$

$$= \frac{1}{2}\sup_{\|q\|_{TV}=1}\int_{s'}\left|\int_s q(ds)\big(K_w(s, ds') - K_{w'}(s, ds')\big)\right|$$

$$\leq \frac{1}{2}\sup_{\|q\|_{TV}=1}\int_{s'}\int_s q(ds)\left|K_w(s, ds') - K_{w'}(s, ds')\right|$$

$$= \frac{1}{2}\sup_{\|q\|_{TV}=1}\int_{s'}\int_s q(ds)\left|\int_a \widetilde{\mathsf{P}}(ds'|s, a)\big(\pi_{w'}(da|s) - \pi_w(da|s)\big)\right|$$

$$\leq \frac{1}{2}\sup_{\|q\|_{TV}=1}\int_s q(ds)\int_a |\pi_{w'}(da|s) - \pi_w(da|s)|\int_{s'}\widetilde{\mathsf{P}}(ds'|s, a)$$

$$= \sup_{\|q\|_{TV}=1}\int_s q(ds)\left\|\pi_{w'}(\cdot|s) - \pi_w(\cdot|s)\right\|_{TV}$$

$$\overset{(i)}{\leq} C_\pi \left\| w' - w \right\|_2 , \tag{16}$$

where $(i)$ follows from Assumption 1. Substituting eq. (16) into eq. (15) yields

$$\left\| P_{\pi_w,I}(\cdot) - P_{\pi_{w'},I}(\cdot) \right\|_{TV} \leq C_\pi \left( \lceil \log_\rho \kappa^{-1} \rceil + \frac{1}{1-\rho} \right) \left\| w' - w \right\|_2 . \tag{17}$$

Then we bound $\left\| \nu_{\pi_w,I}(\cdot,\cdot) - \nu_{\pi_{w'},I}(\cdot,\cdot) \right\|_{TV}$ as follows:

$$
\begin{aligned}
&\left\| \nu_{\pi_w,I}(\cdot,\cdot) - \nu_{\pi_{w'},I}(\cdot,\cdot) \right\|_{TV} \\
&= \left\| P_{\pi_w,I}(\cdot)\pi_w(\cdot|\cdot) - P_{\pi_{w'},I}(\cdot)\pi_{w'}(\cdot|\cdot) \right\|_{TV} \\
&= \frac{1}{2} \int_s \int_a \left| P_{\pi_w,I}(ds)\pi_w(da|s) - P_{\pi_{w'},I}(ds)\pi_{w'}(da|s) \right| \\
&= \frac{1}{2} \int_s \int_a \left| P_{\pi_w,I}(ds)\pi_w(da|s) - P_{\pi_w,I}(ds)\pi_{w'}(da|s) + P_{\pi_w,I}(ds)\pi_{w'}(da|s) - P_{\pi_{w'},I}(ds)\pi_{w'}(da|s) \right| \\
&= \frac{1}{2} \int_s \int_a \left| P_{\pi_w,I}(ds)\pi_w(da|s) - P_{\pi_w,I}(ds)\pi_{w'}(da|s) \right| + \frac{1}{2} \int_s \int_a \left| P_{\pi_w,I}(ds)\pi_{w'}(da|s) - P_{\pi_{w'},I}(ds)\pi_{w'}(da|s) \right| \\
&= \frac{1}{2} \int_s \int_a P_{\pi_w,I}(ds) \left| \pi_w(da|s) - \pi_{w'}(da|s) \right| + \frac{1}{2} \int_s \int_a \left| P_{\pi_w,I}(ds) - P_{\pi_{w'},I}(ds) \right| \pi_{w'}(da|s) \\
&\overset{(i)}{\leq} C_\pi \left\| w - w' \right\|_2 \int_s P_{\pi_w,I}(ds) + \frac{1}{2} \int_s \left| P_{\pi_w,I}(ds) - P_{\pi_{w'},I}(ds) \right| \\
&= C_\pi \left\| w - w' \right\|_2 + \left\| P_{\pi_w,I}(\cdot) - P_{\pi_{w'},I}(\cdot) \right\|_{TV} \\
&\leq C_\pi \left\| w - w' \right\|_2 + C_\pi \left( \lceil \log_\rho \kappa^{-1} \rceil + \frac{1}{1-\rho} \right) \left\| w' - w \right\|_2 \\
&= C_\nu \left\| w' - w \right\|_2 ,
\end{aligned}
$$

where $(i)$ follows from Lemma 1. $\qquad\square$

**Lemma 4.** *Suppose Assumptions 1 and 2 hold, for any $w, w' \in \mathbb{R}^d$ and any state-action pair $(s,a) \in \mathcal{S} \times \mathcal{A}$. We have*

$$\left| Q_{\pi_w}(s,a) - Q_{\pi_{w'}}(s,a) \right| \leq L_Q \left\| w - w' \right\|_2 ,$$

*where $L_Q = \frac{2 r_{\max} C_\nu}{1-\gamma}$.*

*Proof.* By definition, we have $Q_{\pi_w}(s,a) = \frac{1}{1-\gamma} \int_{(\hat{s},\hat{a})} r(\hat{s},\hat{a}) dP^{\pi_w}_{(s,a)}(\hat{s},\hat{a})$, where $P^{\pi_w}_{(s,a)}(\hat{s},\hat{a}) = (1-\gamma) \sum_{t=0}^\infty \gamma^t \mathbb{P}(s_t = \hat{s}, a_t = \hat{a}|s_0 = s, a_0 = a, \pi_w)$ is the state-action visitation distribution of the MDP with policy $\pi_w$ and initialization distribution $P(\cdot|s_0 = s, a_0 = a)$. Thus, $P^{\pi_w}_{(s,a)}(\hat{s},\hat{a})$ is also the state-action stationary distribution of the MDP with policy $\pi_w$ and transition kernel $\widetilde{\mathsf{P}}(\cdot|s,a) = \gamma \mathsf{P}(\cdot|s,a) + (1-\gamma)P(\cdot|s_0 = s, a_0 = a)$. We denote $P^{\pi_w}_s(\hat{s})$ as the state stationary distribution for such a MDP. It then follows that

$$
\begin{aligned}
&\left| Q_{\pi_w}(s,a) - Q_{\pi_{w'}}(s,a) \right| \\
&= \frac{1}{1-\gamma} \left| \int_{(\hat{s},\hat{a})} r(\hat{s},\hat{a}) P^{\pi_w}_{(s,a)}(d\hat{s}, d\hat{a}) - \int_{(\hat{s},\hat{a})} r(\hat{s},\hat{a}) dP^{\pi_{w'}}_{(s,a)}(d\hat{s}, d\hat{a}) \right| \\
&\leq \frac{1}{1-\gamma} \int_{(\hat{s},\hat{a})} r(\hat{s},\hat{a}) \left| P^{\pi_w}_{(s,a)}(d\hat{s}, d\hat{a}) - P^{\pi_{w'}}_{(s,a)}(d\hat{s}, d\hat{a}) \right| \\
&\leq \frac{2 r_{\max}}{1-\gamma} \left\| P^{\pi_w}_{(s,a)}(\cdot,\cdot) - P^{\pi_{w'}}_{(s,a)}(\cdot,\cdot) \right\|_{TV} \\
&\overset{(i)}{\leq} \frac{2 r_{\max} C_\nu}{1-\gamma} \left\| w - w' \right\|_2 ,
\end{aligned}
$$

where $(i)$ follows from Lemma 3. $\qquad\square$

**Lemma 5.** *Suppose Assumptions 1 hold, for $w', w'' \in \mathbb{R}^d$. We have*

$$\left\| \nabla_w \mathbb{E}_{\nu_{\pi^*}} \left[ \log \pi_{w'}(a, s) \right] - \nabla_w \mathbb{E}_{\nu_{\pi^*}} \left[ \log \pi_{w''}(a, s) \right] \right\|_2 \leq L_\psi \left\| w' - w'' \right\|_2.$$

*Proof.* By definition, we obtain

$$\left\| \nabla_w \mathbb{E}_{\nu_{\pi^*}} \left[ \log \pi_{w'}(a, s) \right] - \nabla_w \mathbb{E}_{\nu_{\pi^*}} \left[ \log \pi_{w''}(a, s) \right] \right\|_2$$

$$= \left\| \int_{(s,a)} \psi_{w'}(s, a) \nu_{\pi^*}(ds, da) - \int_{(s,a)} \psi_{w''}(s, a) \nu_{\pi^*}(ds, da) \right\|_2$$

$$\leq \int_{(s,a)} \left\| \psi_{w'}(s, a) - \psi_{w''}(s, a) \right\|_2 \nu_{\pi^*}(ds, da)$$

$$\overset{(i)}{\leq} \int_{(s,a)} L_\psi \left\| w' - w'' \right\|_2 \nu_{\pi^*}(ds, da) = L_\psi \left\| w' - w'' \right\|_2,$$

where $(i)$ follows from Assumption 1. $\qquad\square$

**Lemma 6.** *For any $w \in \mathbb{R}^d$, define $\theta_w^* = (F(w) + \lambda I)^{-1} \nabla J(w)$ and $\theta_w^\dagger = F(w)^\dagger \nabla J(w)$. We have $\left\| \theta_w^* - \theta_w^\dagger \right\|_2 \leq C_r \lambda$, where $0 < C_r < +\infty$ is a constant only depending on the policy class.*

*Proof.* By definition, $F(w) \in \mathbb{R}^{d \times d}$ is a symmetric matrix. Thus, if $rank(F(w)) = k \leq d$, then there exist matrices $\Gamma_w \in \mathbb{R}^{d \times d}$ and $\Lambda_w \in \mathbb{R}^{d \times d}$ such that $F(w) = \Lambda_w^\top \Gamma_w \Lambda_w$, where $\Gamma_w = diag[\lambda_1, \lambda_2, \cdots, \lambda_k, 0, 0, \cdots, 0]$ and $\Lambda_w^\top = [\psi_1, \psi_2, \cdots, \psi_k, \psi_{k+1}, \psi_{k+2}, \cdots, \psi_d]$ is an orthogonal matrices with $\{\psi_1, \psi_2, \cdots, \psi_k\}$ spans over the column space $Col(F(w))$ and $\{\psi_{k+1}, \psi_{k+2}, \cdots, \psi_k\} \perp Col(F(w))$. Without loss of generality, we assume that for all $w$, the linear matrix equation $F(w)x = \nabla J(w)$ has at least one solution $x_w^* \in \mathbb{R}^d$. Then we have

$$\theta_w^* = (F(w) + \lambda I)^{-1} \nabla J(w)$$

$$= (\Lambda_w^\top \Gamma_w \Lambda_w + \lambda I)^{-1} \nabla J(w)$$

$$= \Lambda_w^\top (\Gamma_w + \lambda I)^{-1} \Lambda_w \nabla J(w)$$

$$= \Lambda_w^\top diag\left[ \frac{1}{\lambda_1 + \lambda}, \cdots, \frac{1}{\lambda_k + \lambda}, \frac{1}{\lambda}, \cdots, \frac{1}{\lambda} \right] \Lambda_w \nabla J(w)$$

$$\overset{(i)}{=} \Lambda_w^\top diag\left[ \frac{1}{\lambda_1 + \lambda}, \cdots, \frac{1}{\lambda_k + \lambda}, \frac{1}{\lambda}, \cdots, \frac{1}{\lambda} \right] [\psi_1^\top \nabla J(w), \cdots, \psi_k^\top \nabla J(w), 0, \cdots, 0]^\top$$

$$= \Lambda_w^\top \left[ \frac{1}{\lambda_1 + \lambda} \psi_1^\top \nabla J(w), \cdots, \frac{1}{\lambda_k + \lambda} \psi_k^\top \nabla J(w), 0, \cdots, 0 \right]^\top,$$

where $(i)$ follows from the fact that $\nabla J(w) \in Col(F(w))$ and $\{\psi_{k+1}, \psi_{k+2}, \cdots, \psi_k\} \perp Col(F(w))$. Similarly, we also have

$$\theta_w^\dagger = F(w)^\dagger \nabla J(w)$$

$$= (\Lambda_w^\top \Gamma_w \Lambda_w)^\dagger \nabla J(w)$$

$$= \Lambda_w^\top (\Gamma_w)^\dagger \Lambda_w \nabla J(w)$$

$$= \Lambda_w^\top diag\left[ \frac{1}{\lambda_1}, \cdots, \frac{1}{\lambda_k}, 0, \cdots, 0 \right] \Lambda_w \nabla J(w)$$

$$= \Lambda_w^\top diag\left[ \frac{1}{\lambda_1}, \cdots, \frac{1}{\lambda_k}, 0, \cdots, 0 \right] [\psi_1^\top \nabla J(w), \cdots, \psi_k^\top \nabla J(w), 0, \cdots, 0]^\top$$

$$= \Lambda_w^\top \left[ \frac{1}{\lambda_1} \psi_1^\top \nabla J(w), \cdots, \frac{1}{\lambda_k} \psi_k^\top \nabla J(w), 0, \cdots, 0 \right]^\top.$$

Thus we have

$$\theta_w^* - \theta_w^\dagger = \Lambda_w^\top \left[ \left( \frac{1}{\lambda_1 + \lambda} - \frac{1}{\lambda_1} \right) \psi_1^\top \nabla J(w), \cdots, \left( \frac{1}{\lambda_k + \lambda} - \frac{1}{\lambda_1} \right) \psi_k^\top \nabla J(w), 0, \cdots, 0 \right]^\top$$

$$= -\lambda \Lambda_w^\top \left[ \frac{1}{(\lambda_1 + \lambda)\lambda_1} \psi_1^\top \nabla J(w), \cdots, \frac{1}{(\lambda_k + \lambda)\lambda_k} \psi_k^\top \nabla J(w), 0, \cdots, 0 \right]^\top$$

$$= -\lambda \Lambda_w^\top diag \left[ \frac{1}{(\lambda_1 + \lambda)\lambda_1}, \cdots, \frac{1}{(\lambda_k + \lambda)\lambda_k}, 0, \cdots, 0 \right] \Lambda_w \nabla J(w).$$

We can further obtain

$$\left\| \theta_w^* - \theta_w^\dagger \right\|_2 \leq \frac{\lambda}{\lambda_{\min}^2} \|\Lambda_w\|_2^2 \|\nabla J(w)\|_2 \overset{(i)}{\leq} \frac{C_\psi r_{\max}}{\lambda_{\min}^2 (1-\gamma)} \lambda = C_r \lambda,$$

where in $(i)$ we define $\lambda_{\min} = \min_{w \in \mathbb{R}^d} \min_{1 \leq i \leq k_w} \lambda_{w,i}$, with $\lambda_{w,i}$ being the $i$-th element in $\Gamma_w$ and $k_w$ being the rank of the matrix $F(w)$. $\qquad \square$

# F    Proof of Theorem 2

In this section and next section, we assume $C_\psi = 1$ without loss of generality. We restate Theorem 2 as follows to include the specifics of the parameters.

**Theorem 5** (Restatement of Theorem 2). *Consider the AC algorithm in Algorithm 1. Suppose Assumptions 1 and 2 hold, and let the stepsize $\alpha = \frac{1}{4L_J}$. We have*

$$\mathbb{E}[\| \nabla_w J(w_{\hat{T}}) \|_2^2]$$
$$\leq \frac{16 L_J r_{\max}}{(1-\gamma)T} + 18 \frac{\sum_{t=0}^{T-1} \mathbb{E}[\| \theta_t - \theta_{w_t}^* \|_2^2]}{T} + \frac{72(r_{\max} + 2R_\theta)^2 [1 + (\kappa - 1)\rho]}{B(1-\rho)} + C_1 \zeta_{approx}^{critic},$$

*where $C_1$ is a positive constant. Furthermore, let $B \geq \frac{216(r_{\max} + 2R_\theta)^2 [1 + (\kappa - 1)\rho]}{(1-\rho)\epsilon}$ and $T \geq \frac{48 L_J r_{\max}}{(1-\gamma)\epsilon}$. Suppose the same setting of Theorem 1 holds (with $M$ and $T_c$ defined therein) so that $\mathbb{E}\left[ \| \theta_t - \theta_{w_t}^* \|_2^2 \right] \leq \frac{\epsilon}{108}$ for all $0 \leq t \leq T-1$. We have*

$$\mathbb{E}[\| \nabla_w J(w_{\hat{T}}) \|_2^2] \leq \epsilon + \mathcal{O}(\zeta_{approx}^{critic}),$$

*with the total sample complexity given by $(B + MT_c)T = \mathcal{O}((1-\gamma)^{-2} \epsilon^{-2} \log(1/\epsilon))$.*

*Proof.* For brevity, we define $v_t(\theta) = \frac{1}{B} \sum_{i=0}^{B-1} \delta_\theta(s_{t,i}, a_{t,i}, s_{t,i+1}) \psi_{w_t}(s_{t,i}, a_{t,i})$, $A_\theta(s,a) = \mathbb{E}_{\widetilde{\mathsf{P}}}[\delta_\theta(s,a,s')|(s,a)]$, and $g(\theta, w) = \mathbb{E}_{\nu_w}[A_\theta(s,a)\psi_w(s,a)]$ for all $w \in \mathbb{R}^{d_1}$, $\theta \in \mathbb{R}^{d_2}$ and $t \geq 0$. Following from the $L_J$-Lipschitz condition indicated in Proposition 1, we have

$$J(w_{t+1}) \geq J(w_t) + \langle \nabla_w J(w_t), w_{t+1} - w_t \rangle - \frac{L_J}{2} \|w_{t+1} - w_t\|_2^2$$

$$= J(w_t) + \alpha \langle \nabla_w J(w_t), v_t(\theta_t) - \nabla_w J(w_t) + \nabla_w J(w_t) \rangle - \frac{L_J \alpha^2}{2} \|v_t(\theta_t)\|_2^2$$

$$= J(w_t) + \alpha \|\nabla_w J(w_t)\|_2^2 + \alpha \langle \nabla_w J(w_t), v_t - \nabla_w J(w_t) \rangle$$
$$\quad - \frac{L_J \alpha^2}{2} \|v_t(\theta_t) - \nabla_w J(w_t) + \nabla_w J(w_t)\|_2^2$$

$$\overset{(i)}{\geq} J(w_t) + \left( \frac{1}{2}\alpha - L_J \alpha^2 \right) \|\nabla_w J(w_t)\|_2^2 - \left( \frac{1}{2}\alpha + L_J \alpha^2 \right) \|v_t(\theta_t) - \nabla_w J(w_t)\|_2^2, \tag{18}$$

where $(i)$ follows because

$$\langle \nabla_w J(w_t), v_t(\theta_t) - \nabla_w J(w_t) \rangle \geq -\frac{1}{2} \|\nabla_w J(w_t)\|_2^2 - \frac{1}{2} \|v_t(\theta_t) - \nabla_w J(w_t)\|_2^2,$$

and

$$\|v_t(\theta_t) - \nabla_w J(w_t) + \nabla_w J(w_t)\|_2^2 \leq 2 \|v_t(\theta_t) - \nabla_w J(w_t)\|_2^2 + 2 \|\nabla_w J(w_t)\|_2^2.$$

Taking expectation on both sides of eq. (18) conditioned on $\mathcal{F}_t$ and rearranging eq. (18) yield

$$\left( \frac{1}{2}\alpha - L_J \alpha^2 \right) \mathbb{E}[\| \nabla_w J(w_t) \|_2^2 | \mathcal{F}_t]$$

$$\leq \mathbb{E}[J(w_{t+1})|\mathcal{F}_t] - J(w_t) + \left(\frac{1}{2}\alpha + L_J\alpha^2\right)\mathbb{E}[\|v_t(\theta_t) - \nabla_w J(w_t)\|_2^2 \,|\mathcal{F}_t]. \quad (19)$$

Then, we upper-bound the term $\mathbb{E}[\|v_t(\theta_t) - \nabla_w J(w_t)\|_2^2 \,|\mathcal{F}_t]$ as follows. By definition, we have

$$\|v_t(\theta_t) - \nabla_w J(w_t)\|_2^2$$
$$= \left\|v_t(\theta_t) - v_t(\theta_{w_t}^*) + v_t(\theta_{w_t}^*) - g(\theta_{w_t}^*, w_t) + g(\theta_{w_t}^*, w_t) - \nabla_w J(w_t)\right\|_2^2$$
$$\leq 3\left\|v_t(\theta_t) - v_t(\theta_{w_t}^*)\right\|_2^2 + 3\left\|v_t(\theta_{w_t}^*) - g(\theta_{w_t}^*, w_t)\right\|_2^2 + 3\left\|g(\theta_{w_t}^*, w_t) - \nabla_w J(w_t)\right\|_2^2, \quad (20)$$

in which

$$\left\|v_t(\theta_t) - v_t(\theta_{w_t}^*)\right\|_2^2$$
$$= \left\|\frac{1}{B}\sum_{i=0}^{B-1}\left[\delta_{\theta_t}(s_{t,i}, a_{t,i}, s_{t,i+1}) - \delta_{\theta_{w_t}^*}(s_{t,i}, a_{t,i}, s_{t,i+1})\right]\psi_{w_t}(s_{t,i}, a_{t,i})\right\|_2^2$$
$$\leq \frac{1}{B}\sum_{i=0}^{B-1}\left\|\left[\delta_{\theta_t}(s_{t,i}, a_{t,i}, s_{t,i+1}) - \delta_{\theta_{w_t}^*}(s_{t,i}, a_{t,i}, s_{t,i+1})\right]\psi_{w_t}(s_{t,i}, a_{t,i})\right\|_2^2$$
$$\leq \frac{1}{B}\sum_{i=0}^{B-1}\left\|\delta_{\theta_t}(s_{t,i}, a_{t,i}, s_{t,i+1}) - \delta_{\theta_{w_t}^*}(s_{t,i}, a_{t,i}, s_{t,i+1})\right\|_2^2$$
$$= \frac{1}{B}\sum_{i=0}^{B-1}\left\|\gamma(V_{\theta_t}(s_{t,i+1}) - V_{\theta_{w_t}^*}(s_{t,i+1})) + (V_{\theta_{w_t}^*}(s_{t,i}) - V_{\theta_t}(s_{t,i}))\right\|_2^2$$
$$= \frac{1}{B}\sum_{i=0}^{B-1}\left\|(\gamma\phi(s_{t,i+1}) - \phi(s_{t,i}))^\top(\theta_t - \theta_{w_t}^*)\right\|_2^2 \leq 4\left\|\theta_t - \theta_{w_t}^*\right\|_2^2, \quad (21)$$

and

$$\left\|g(\theta_{w_t}^*, w_t) - \nabla_w J(w_t)\right\|_2^2$$
$$= \left\|\mathbb{E}_{\nu_{w_t}}[A_{\theta_{w_t}^*}(s, a)\psi_{w_t}(s, a)] - \mathbb{E}_{\nu_{w_t}}[A_{\pi_{w_t}}(s, a)\psi_{w_t}(s, a)]\right\|_2^2$$
$$= \left\|\mathbb{E}_{\nu_{w_t}}\left[\left(A_{\theta_{w_t}^*}(s, a) - A_{\pi_{w_t}}(s, a)\right)\psi_{w_t}(s, a)\right]\right\|_2^2$$
$$\leq \mathbb{E}_{\nu_{w_t}}\left[\left\|\left(A_{\theta_{w_t}^*}(s, a) - A_{\pi_{w_t}}(s, a)\right)\psi_{w_t}(s, a)\right\|_2^2\right] \leq \mathbb{E}_{\nu_{w_t}}\left[\left\|A_{\theta_{w_t}^*}(s, a) - A_{\pi_{w_t}}(s, a)\right\|_2^2\right]$$
$$= \mathbb{E}_{\nu_{w_t}}\left[\left|\gamma\mathbb{E}\left[V_{\theta_{w_t}^*}(s') - V_{\pi_{w_t}}(s')\big|(s, a)\right] + V_{\pi_{w_t}}(s) - V_{\theta_{w_t}^*}(s)\right|^2\right]$$
$$\leq 2\mathbb{E}_{\nu_{w_t}}\left[\left|\gamma\mathbb{E}\left[V_{\theta_{w_t}^*}(s') - V_{\pi_{w_t}}(s')\big|(s, a)\right]\right|^2\right] + 2\mathbb{E}\left[\left|V_{\pi_{w_t}}(s) - V_{\theta_{w_t}^*}(s)\right|^2\right]$$
$$\overset{(i)}{\leq} 4\zeta_{\text{approx}}^{\text{critic}}, \quad (22)$$

where $(i)$ follows from the definition $\zeta_{\text{approx}}^{\text{critic}} = \max_{w\in\mathcal{W}}\mathbb{E}_{\nu_w}[|V_{\pi_w}(s) - V_{\theta_{\pi_w}^*}(s)|^2]$. Substituting eq. (21) and eq. (22) into eq. (20) yields

$$\mathbb{E}[\|v_t(\theta_t) - \nabla_w J(w_t)\|_2^2 \,|\mathcal{F}_t]$$
$$\leq 3\mathbb{E}\left[\left\|v_t(\theta_{w_t}^*) - g(\theta_{w_t}^*, w_t)\right\|_2^2 \,|\mathcal{F}_t\right] + 12\left\|\theta_t - \theta_{w_t}^*\right\|_2^2 + 12\zeta_{\text{approx}}^{\text{critic}}. \quad (23)$$

To upper bound the first term on the right-hand-side of eq. (23), we proceed as follows.

$$\mathbb{E}\left[\left\|v_t(\theta_{w_t}^*) - g(\theta_{w_t}^*, w_t)\right\|_2^2 \,|\mathcal{F}_t\right]$$
$$= \mathbb{E}\left[\left\|\frac{1}{B}\sum_{i=0}^{B-1}\delta_{\theta_{w_t}^*}(s_{t,i}, a_{t,i}, s_{t,i+1})\psi_{w_t}(s_{t,i}, a_{t,i}) - \mathbb{E}_{\nu_w}\left[A_{\theta_{w_t}^*}(s, a)\psi_{w_t}(s, a)\right]\right\|_2^2 \,\bigg|\mathcal{F}_t\right]$$

$$= \frac{1}{B^2} \sum_{i=0}^{B-1} \sum_{j=0}^{B-1} \mathbb{E}\Big[\Big\langle \delta_{\theta_{w_t}^*}(s_{t,i},a_{t,i},s_{t,i+1})\psi_{w_t}(s_{t,i},a_{t,i}) - \mathbb{E}_{\nu_w}\Big[A_{\theta_{w_t}^*}(s,a)\psi_{w_t}(s,a)\Big],$$

$$\delta_{\theta_{w_t}^*}(s_{t,j},a_{t,j},s_{t,j+1})\psi_{w_t}(s_{t,j},a_{t,j}) - \mathbb{E}_{\nu_w}\Big[A_{\theta_{w_t}^*}(s,a)\psi_{w_t}(s,a)\Big]\Big\rangle\Big|\mathcal{F}_t\Big]$$

$$\overset{(i)}{\leq} \frac{1}{B^2}\Bigg[4B\left(r_{\max}+2R_\theta\right)^2$$

$$+ \sum_{i\neq j}\mathbb{E}\Big[\Big\langle \delta_{\theta_{w_t}^*}(s_{t,i},a_{t,i},s_{t,i+1})\psi_{w_t}(s_{t,i},a_{t,i}) - \mathbb{E}_{\nu_w}\Big[A_{\theta_{w_t}^*}(s,a)\psi_{w_t}(s,a)\Big],$$

$$\delta_{\theta_{w_t}^*}(s_{t,j},a_{t,j},s_{t,j+1})\psi_{w_t}(s_{t,j},a_{t,j}) - \mathbb{E}_{\nu_w}\Big[A_{\theta_{w_t}^*}(s,a)\psi_{w_t}(s,a)\Big]\Big\rangle\Big|\mathcal{F}_t\Big]\Bigg], \quad (24)$$

where $(i)$ follows from the fact that $\Big|\delta_{\theta_{w_t}^*}(s_{t,j},a_{t,j},s_{t,j+1})\psi_{w_t}(s_{t,j},a_{t,j})\Big| \leq r_{\max}+2R_\theta$ and $\Big|\mathbb{E}_{\nu_w}[A_{\theta_{w_t}^*}(s,a)\psi_{w_t}(s,a)]\Big| \leq r_{\max}+2R_\theta$. We next upper bound the following term for the case $i>j$.

$$\mathbb{E}\Big[\Big\langle \delta_{\theta_{w_t}^*}(s_{t,i},a_{t,i},s_{t,i+1})\psi_{w_t}(s_{t,i},a_{t,i}) - \mathbb{E}_{\nu_w}\Big[A_{\theta_{w_t}^*}(s,a)\psi_{w_t}(s,a)\Big],$$

$$\delta_{\theta_{w_t}^*}(s_{t,j},a_{t,j},s_{t,j+1})\psi_{w_t}(s_{t,j},a_{t,j}) - \mathbb{E}_{\nu_w}\Big[A_{\theta_{w_t}^*}(s,a)\psi_{w_t}(s,a)\Big]\Big\rangle\Big|\mathcal{F}_t\Big]$$

$$= \mathbb{E}\Big[\mathbb{E}\Big[\Big\langle \delta_{\theta_{w_t}^*}(s_{t,i},a_{t,i},s_{t,i+1})\psi_{w_t}(s_{t,i},a_{t,i}) - \mathbb{E}_{\nu_w}\Big[A_{\theta_{w_t}^*}(s,a)\psi_{w_t}(s,a)\Big],$$

$$\delta_{\theta_{w_t}^*}(s_{t,j},a_{t,j},s_{t,j+1})\psi_{w_t}(s_{t,j},a_{t,j}) - \mathbb{E}_{\nu_w}\Big[A_{\theta_{w_t}^*}(s,a)\psi_{w_t}(s,a)\Big]\Big\rangle\Big|\mathcal{F}_{t,j}\Big]\Big|\mathcal{F}_t\Big]$$

$$= \mathbb{E}\Big[\Big\langle \mathbb{E}\Big[\delta_{\theta_{w_t}^*}(s_{t,i},a_{t,i},s_{t,i+1})\psi_{w_t}(s_{t,i},a_{t,i})\Big|\mathcal{F}_{t,j}\Big] - \mathbb{E}_{\nu_w}\Big[A_{\theta_{w_t}^*}(s,a)\psi_{w_t}(s,a)\Big],$$

$$\delta_{\theta_{w_t}^*}(s_{t,j},a_{t,j},s_{t,j+1})\psi_{w_t}(s_{t,j},a_{t,j}) - \mathbb{E}_{\nu_w}\Big[A_{\theta_{w_t}^*}(s,a)\psi_{w_t}(s,a)\Big]\Big\rangle\Big]\Big|\mathcal{F}_t\Big]$$

$$= \mathbb{E}\Big[\Big\langle \mathbb{E}\Big[A_{\theta_{w_t}^*}(s_{t,i},a_{t,i})\psi_{w_t}(s_{t,i},a_{t,i})\Big|\mathcal{F}_{t,j}\Big] - \mathbb{E}_{\nu_w}\Big[A_{\theta_{w_t}^*}(s,a)\psi_{w_t}(s,a)\Big],$$

$$\delta_{\theta_{w_t}^*}(s_{t,j},a_{t,j},s_{t,j+1})\psi_{w_t}(s_{t,j},a_{t,j}) - \mathbb{E}_{\nu_w}\Big[A_{\theta_{w_t}^*}(s,a)\psi_{w_t}(s,a)\Big]\Big\rangle\Big]\Big|\mathcal{F}_t\Big]$$

$$\leq \mathbb{E}\Big[\Big\|\mathbb{E}\Big[A_{\theta_{w_t}^*}(s_{t,i},a_{t,i})\psi_{w_t}(s_{t,i},a_{t,i})\Big|\mathcal{F}_{t,j}\Big] - \mathbb{E}_{\nu_w}\Big[A_{\theta_{w_t}^*}(s,a)\psi_{w_t}(s,a)\Big]\Big\|_2$$

$$\Big\|\delta_{\theta_{w_t}^*}(s_{t,j},a_{t,j},s_{t,j+1})\psi_{w_t}(s_{t,j},a_{t,j}) - \mathbb{E}_{\nu_w}\Big[A_{\theta_{w_t}^*}(s,a)\psi_{w_t}(s,a)\Big]\Big\|_2\Big|\mathcal{F}_t\Big]$$

$$\leq 2(r_{\max}+2R_\theta)\mathbb{E}\Big[\Big\|\mathbb{E}\Big[A_{\theta_{w_t}^*}(s_{t,i},a_{t,i})\psi_{w_t}(s_{t,i},a_{t,i})\Big|\mathcal{F}_{t,j}\Big] - \mathbb{E}_{\nu_w}\Big[A_{\theta_{w_t}^*}(s,a)\psi_{w_t}(s,a)\Big]\Big\|_2\Big|\mathcal{F}_t\Big]$$

$$\overset{(i)}{\leq} 4(r_{\max}+2R_\theta)^2\kappa\rho^{i-j},$$

where $(i)$ follows from Assumption 2 and the fact that

$$\Big\|\mathbb{E}\Big[A_{\theta_{w_t}^*}(s_{t,i},a_{t,i})\psi_{w_t}(s_{t,i},a_{t,i})\Big|\mathcal{F}_{t,j}\Big] - \mathbb{E}_{\nu_w}\Big[A_{\theta_{w_t}^*}(s,a)\psi_{w_t}(s,a)\Big]\Big\|_2$$

$$= \Big\|\int_{x_{t,i}} A_{\theta_{w_t}^*}(x_{t,i})\psi_{w_t}(x_{t,i})P(dx_{t,i}|\mathcal{F}_{t,j}) - \int_{x_{t,i}} A_{\theta_{w_t}^*}(x_{t,i})\psi_{w_t}(x_{t,i})\nu_{\pi_{w_t}}(dx_{t,i})\Big\|_2$$

$$\leq \int_{x_i} \Big\|A_{\theta_{w_t}^*}(x_{t,i})\psi_{w_t}(x_{t,i})\Big\|_2 |P(dx_{t,i}|\mathcal{F}_{t,j}) - \nu_{\pi_{w_t}}(dx_{t,i})|$$

$$\leq 2(r_{\max}+2R_\theta)\Big\|P(\cdot|\mathcal{F}_{t,j}) - \nu_{\pi_{w_t}}(\cdot)\Big\|_{TV} \leq 2(r_{\max}+2R_\theta)\kappa\rho^{i-j}, \quad (25)$$

where we denote $x_{t,k} = (s_{t,k},a_{t,k})$ for $k \geq 0$ for convenience. Substituting eq. (25) into eq. (24) yields

$$\mathbb{E}\Big[\big\|v_t(\theta_{w_t}^*) - g(\theta_{w_t}^*,w_t)\big\|_2^2 \big|\mathcal{F}_t\Big] \leq \frac{1}{B^2}\Bigg[4B\left(r_{\max}+2R_\theta\right)^2 + 4(r_{\max}+2R_\theta)^2\kappa\sum_{i\neq j}\rho^{i-j}\Bigg]$$

$$\leq \frac{1}{B^2} \left[ 4B \left( r_{\max} + 2R_\theta \right)^2 + \frac{8(r_{\max} + 2R_\theta)^2 \kappa \rho B}{1 - \rho} \right]$$

$$\leq \frac{8(r_{\max} + 2R_\theta)^2 [1 + (\kappa - 1)\rho]}{B(1 - \rho)}. \tag{26}$$

Substituting eq. (26) into eq. (23) yields

$$\mathbb{E}[\|v_t(\theta_t) - \nabla_w J(w_t)\|_2^2 \,|\, \mathcal{F}_t]$$
$$\leq \frac{24(r_{\max} + 2R_\theta)^2 [1 + (\kappa - 1)\rho]}{B(1 - \rho)} + 12 \left\| \theta_t - \theta_{w_t}^* \right\|_2^2 + 12\zeta_{\text{approx}}^{\text{critic}}. \tag{27}$$

Then, substituting eq. (27) into eq. (19) and taking expectation of $\mathcal{F}_t$ on both sides yield

$$\left( \frac{1}{2}\alpha - L_J \alpha^2 \right) \mathbb{E}[\|\nabla_w J(w_t)\|_2^2]$$

$$\leq \mathbb{E}[J(w_{t+1})] - \mathbb{E}[J(w_t)] + 12 \left( \frac{1}{2}\alpha + L_J \alpha^2 \right) \mathbb{E}[\left\| \theta_t - \theta_{w_t}^* \right\|_2^2] + 12 \left( \frac{1}{2}\alpha + L_J \alpha^2 \right) \zeta_{\text{approx}}^{\text{critic}}$$

$$+ 24 \left( \frac{1}{2}\alpha + L_J \alpha^2 \right) \frac{(r_{\max} + 2R_\theta)^2 [1 + (\kappa - 1)\rho]}{B(1 - \rho)}. \tag{28}$$

Letting $\alpha = \frac{1}{4L_J}$ and dividing both sides of eq. (28) by $1/(16L_J)$ yield

$$\mathbb{E}[\|\nabla_w J(w_t)\|_2^2] \leq 16 L_J \left( \mathbb{E}[J(w_{t+1})] - \mathbb{E}[J(w_t)] \right) + 36 \mathbb{E}[\left\| \theta_t - \theta_{w_t}^* \right\|_2^2] + 36 \zeta_{\text{approx}}^{\text{critic}}$$

$$+ \frac{72(r_{\max} + 2R_\theta)^2 [1 + (\kappa - 1)\rho]}{B(1 - \rho)}. \tag{29}$$

Taking the summation of eq. (29) over $t = \{0, \cdots, T - 1\}$ and dividing both sides by $T$ yield

$$\mathbb{E}[\left\| \nabla_w J(w_{\hat{T}}) \right\|_2^2] = \frac{1}{T} \sum_{t=0}^{T-1} \mathbb{E}[\|\nabla_w J(w_t)\|_2^2]$$

$$\leq \frac{16 L_J \left( \mathbb{E}[J(w_T)] - J(w_0) \right)}{T} + 36 \frac{\sum_{t=0}^{T-1} \mathbb{E}[\left\| \theta_t - \theta_{w_t}^* \right\|_2^2]}{T}$$

$$+ \frac{72(r_{\max} + 2R_\theta)^2 [1 + (\kappa - 1)\rho]}{B(1 - \rho)} + C_1 \zeta_{\text{approx}}^{\text{critic}}$$

$$\leq \frac{16 L_J r_{\max}}{(1 - \gamma)T} + 36 \frac{\sum_{t=0}^{T-1} \mathbb{E}[\left\| \theta_t - \theta_{w_t}^* \right\|_2^2]}{T}$$

$$+ \frac{72(r_{\max} + 2R_\theta)^2 [1 + (\kappa - 1)\rho]}{B(1 - \rho)} + C_1 \zeta_{\text{approx}}^{\text{critic}}. \tag{30}$$

Letting $B \geq \frac{216(r_{\max} + 2R_\theta)^2 [1 + (\kappa - 1)\rho]}{(1 - \rho)\epsilon}$, $\mathbb{E}\left[ \left\| \theta_t - \theta_{w_t}^* \right\|_2^2 \right] \leq \frac{\epsilon}{108}$ for all $0 \leq t \leq T - 1$, and $T \geq \frac{48 L_J r_{\max}}{(1 - \gamma)\epsilon}$, then we have

$$\frac{1}{T} \sum_{t=0}^{T-1} \mathbb{E}[\|\nabla_w J(w_i)\|_2^2] \leq \epsilon + \mathcal{O}(\zeta_{\text{approx}}^{\text{critic}}).$$

The total sample complexity is given by

$$(B + MT_c)T = \mathcal{O}\left[ \left( \frac{1}{\epsilon} + \frac{1}{\epsilon} \log\left( \frac{1}{\epsilon} \right) \right) \frac{1}{(1 - \gamma)^2 \epsilon} \right] = \mathcal{O}\left( \frac{1}{(1 - \gamma)^2 \epsilon^2} \log\left( \frac{1}{\epsilon} \right) \right).$$

$\square$

# G  Proof of Theorem 3

We restate Theorem 3 as follows to include the specifics of the parameters.

**Theorem 6** (Restatement of Theorem 3). *Consider the NAC algorithm in Algorithm 1. Suppose Assumptions 1 and 2 hold, and let the stepsize $\alpha = \frac{\lambda^2}{4L_J(1+\lambda)}$. We have*

$$J(\pi^*) - \mathbb{E}\big[J(\pi_{w_{\hat{T}}})\big] \leq \frac{4L_J(1+\lambda)(D(w_0) - \mathbb{E}[D(w_T)])}{T(1-\gamma)\lambda^2} + \frac{4L_\psi(1+\lambda)}{\lambda^2(1-\gamma)}\frac{\mathbb{E}[J(w_T)] - J(w_0)}{T}$$

$$+ \frac{81L_\psi(1+\lambda)}{\lambda^2(1-\gamma)L_J}\frac{\sum_{t=0}^{T-1}\mathbb{E}\big[\|\theta_t - \theta_{w_t}^*\|_2^2\big]}{T}$$

$$+ \frac{3L_\psi(1+\lambda)}{(1-\gamma)L_J}\left(\frac{8r_{\max}^2}{\lambda^4(1-\gamma)^2} + \frac{108(r_{\max}+2R_\theta)^2}{\lambda^2}\right)\frac{1+(\kappa-1)\rho}{(1-\rho)B}$$

$$+ \frac{162L_\psi(1+\lambda)}{\lambda^2(1-\gamma)L_J}\zeta_{approx}^{critic} + \frac{16\sqrt{\zeta_{approx}^{critic}}}{\lambda(1-\gamma)} + \frac{64L_\psi\zeta_{approx}^{critic}}{(1-\gamma)L_J(1+\lambda)}$$

$$+ \sqrt{\frac{1}{(1-\gamma)^3}}\left\|\frac{\nu_{\pi^*}}{\nu_{\pi_{w_0}}}\right\|_\infty \sqrt{\zeta_{approx}^{actor}} + \frac{C_r\lambda}{1-\gamma}, \tag{31}$$

*where $\lambda$ is the regularizing coefficient for estimating the inverse of Fisher information matrix. Furthermore, let*

$$T \geq \max\left\{\frac{16L_J(1+\lambda)}{\epsilon(1-\gamma)\lambda^2}, \frac{16r_{\max}L_\psi(1+\lambda)}{\epsilon(1-\gamma)^2\lambda^2}\right\},$$

$$B \geq \max\left\{\frac{24(r_{\max}+2R_\theta)^2[1+(\kappa-1)\rho]}{(1-\rho)\zeta_{approx}^{critic}}, \frac{8r_{\max}^2[1+(\kappa-1)\rho]}{\lambda^2(1-\gamma)^2(1-\rho)\zeta_{approx}^{critic}}, \right.$$

$$\left. \frac{3L_\psi(1+\lambda)}{\epsilon(1-\gamma)L_J}\left(\frac{32r_{\max}^2}{\lambda^4(1-\gamma)^2} + \frac{432(r_{\max}+2R_\theta)^2}{\lambda^2}\right)\frac{1+(\kappa-1)\rho}{(1-\rho)}\right\},$$

$$\lambda = \sqrt{\zeta_{approx}^{critic}}.$$

*Suppose the same setting of Theorem 1 holds (with $M$ and $T_c$ defined therein) so that*

$$\mathbb{E}\left[\|\theta_t - \theta_{w_t}^*\|_2^2\right] \leq \min\left\{\frac{\zeta_{approx}^{critic}}{64}, \frac{\epsilon\lambda^2(1-\gamma)L_J}{324L_\psi(1+\lambda)}\right\}, \quad \text{for all} \quad 0 \geq t \geq T-1.$$

*We have*

$$J(\pi^*) - \frac{1}{T}\sum_{t=0}^{T-1}\mathbb{E}[J(\pi_{w_t})] \leq \epsilon + \mathcal{O}\left(\frac{\sqrt{\zeta_{approx}^{actor}}}{(1-\gamma)^{1.5}}\right) + \mathcal{O}\left(\frac{\sqrt{\zeta_{approx}^{critic}}}{1-\gamma}\right),$$

*with the total sample complexity given by $(B + MT_c)T = \mathcal{O}((1-\gamma)^{-4}\epsilon^{-2}\log(1/\epsilon))$.*

*Proof.* We first show that NAC in Algorithm 1 convergences to a neighbourhood of a first-order stationary point. Then we present the proof of Theorem 3/Theorem 6, in which the convergence of NAC is characterized in terms of the function value.

Recall the definition of $v_t(\theta)$ in Appendix F, we define

$$u_t(\theta) = \big[F_t(w_t) + \lambda I\big]^{-1}\Big[\frac{1}{B}\sum_{i=0}^{B-1}\delta_\theta(s_{t,i}, a_{t,i})\psi_{w_t}(s_{t,i}, a_{t,i}, s_{t,i+1})\Big] = \big[F_t(w_t) + \lambda I\big]^{-1}v_t(\theta).$$

Following from the $L_J$-Lipschitz condition indicated in Proposition 1, we have

$$J(w_{t+1}) \geq J(w_t) + \langle\nabla_w J(w_t), w_{t+1} - w_t\rangle - \frac{L_J}{2}\|w_{t+1} - w_t\|_2^2$$

$$= J(w_t) + \alpha\langle\nabla_w J(w_t), u_t(\theta_t)\rangle - \frac{L_J\alpha^2}{2}\|u_t(\theta_t)\|_2^2$$

$$= J(w_t) + \alpha\langle\nabla_w J(w_t), (F(w_t) + \lambda I)^{-1}\nabla_w J(w_t)\rangle$$

$$+ \alpha\langle\nabla_w J(w_t), u_t(\theta_t) - (F(w_t) + \lambda I)^{-1}\nabla_w J(w_t)\rangle$$

$$- \frac{L_J \alpha^2}{2} \left\| u_t(\theta_t) - (F(w_t) + \lambda I)^{-1} \nabla_w J(w_t) + (F(w_t) + \lambda I)^{-1} \nabla_w J(w_t) \right\|_2^2$$

$$\overset{(i)}{\geq} J(w_t) + \frac{\alpha}{1 + \lambda} \left\| \nabla_w J(w_t) \right\|_2^2 + \alpha \langle \nabla_w J(w_t), u_t(\theta_t) - (F(w_t) + \lambda I)^{-1} \nabla_w J(w_t) \rangle$$

$$- L_J \alpha^2 \left\| u_t(\theta_t) - (F(w_t) + \lambda I)^{-1} \nabla_w J(w_t) \right\|_2^2 - L_J \alpha^2 \left\| (F(w_t) + \lambda I)^{-1} \nabla_w J(w_t) \right\|_2^2$$

$$\overset{(ii)}{\geq} J(w_t) + \frac{\alpha}{1 + \lambda} \left\| \nabla_w J(w_t) \right\|_2^2$$

$$- \alpha \left( \frac{1}{2(1 + \lambda)} \left\| \nabla_w J(w_t) \right\|_2^2 + \frac{1 + \lambda}{2} \left\| u_t(\theta_t) - (F(w_t) + \lambda I)^{-1} \nabla_w J(w_t) \right\|_2^2 \right)$$

$$- L_J \alpha^2 \left\| u_t(\theta_t) - (F(w_t) + \lambda I)^{-1} \nabla_w J(w_t) \right\|_2^2 - \frac{L_J \alpha^2}{\lambda^2} \left\| \nabla_w J(w_t) \right\|_2^2$$

$$= J(w_t) + \left( \frac{\alpha}{2(1 + \lambda)} - \frac{L_J \alpha^2}{\lambda^2} \right) \left\| \nabla_w J(w_t) \right\|_2^2$$

$$- \left( \frac{\alpha(1 + \lambda)}{2} + L_J \alpha^2 \right) \left\| u_t(\theta_t) - (F(w_t) + \lambda I)^{-1} \nabla_w J(w_t) \right\|_2^2, \tag{32}$$

where $(i)$ follows because $\langle \nabla_w J(w_t), (F(w_t) + \lambda I)^{-1} \nabla_w J(w_t) \rangle \geq \frac{1}{1+\lambda} \left\| \nabla_w J(w_t) \right\|_2^2$, and $(ii)$ follows from the fact that $\left\| (F(w_t) + \lambda I)^{-1} \nabla_w J(w_t) \right\|_2^2 \leq \frac{1}{\lambda^2} \left\| \nabla_w J(w_t) \right\|_2^2$ and Young's inequality. To bound the term $\left\| u_t(\theta_t) - (F(w_t) + \lambda I)^{-1} \nabla_w J(w_t) \right\|_2^2$, we proceed as follows:

$$\left\| u_t(\theta_t) - (F(w_t) + \lambda I)^{-1} \nabla_w J(w_t) \right\|_2^2$$

$$= \left\| u_t(\theta_t) - (F(w_t) + \lambda I)^{-1} v_t(\theta_t) + (F(w_t) + \lambda I)^{-1} v_t(\theta_t) - (F(w_t) + \lambda I)^{-1} \nabla_w J(w_t) \right\|_2^2$$

$$\leq 2 \left\| u_t(\theta_t) - (F(w_t) + \lambda I)^{-1} v_t(\theta_t) \right\|_2^2 + 2 \left\| (F(w_t) + \lambda I)^{-1} v_t(\theta_t) - (F(w_t) + \lambda I)^{-1} \nabla_w J(w_t) \right\|_2^2$$

$$= 2 \left\| \left[ (F_t(w_t) + \lambda I)^{-1} - (F(w_t) + \lambda I)^{-1} \right] v_t(\theta_t) \right\|_2^2 + 2 \left\| (F(w_t) + \lambda I)^{-1} (v_t(\theta_t) - \nabla_w J(w_t)) \right\|_2^2$$

$$= 2 \left\| \left[ (F_t(w_t) + \lambda I)^{-1} - (F(w_t) + \lambda I)^{-1} \right] (v_t(\theta_t) - \nabla_w J(w_t) + \nabla_w J(w_t)) \right\|_2^2$$

$$+ 2 \left\| (F(w_t) + \lambda I)^{-1} (v_t - \nabla_w J(w_t)) \right\|_2^2$$

$$\leq 4 \left\| \left[ (F_t(w_t) + \lambda I)^{-1} - (F(w_t) + \lambda I)^{-1} \right] (v_t(\theta_t) - \nabla_w J(w_t)) \right\|_2^2$$

$$+ 2 \left\| (F(w_t) + \lambda I)^{-1} (v_t - \nabla_w J(w_t)) \right\|_2^2$$

$$+ 4 \left\| \left[ (F_t(w_t) + \lambda I)^{-1} - (F(w_t) + \lambda I)^{-1} \right] \nabla_w J(w_t) \right\|_2^2$$

$$\leq \left[ 4 \left\| (F_t(w_t) + \lambda I)^{-1} - (F(w_t) + \lambda I)^{-1} \right\|_2^2 + 2 \left\| (F(w_t) + \lambda I)^{-1} \right\|_2^2 \right] \left\| v_t(\theta_t) - \nabla_w J(w_t) \right\|_2^2$$

$$+ 4 \left\| (F_t(w_t) + \lambda I)^{-1} - (F(w_t) + \lambda I)^{-1} \right\|_2^2 \left\| \nabla_w J(w_t) \right\|_2^2$$

$$\leq \left[ 8 \left\| (F_t(w_t) + \lambda I)^{-1} \right\|_2^2 + 10 \left\| (F(w_t) + \lambda I)^{-1} \right\|_2^2 \right] \left\| v_t(\theta_t) - \nabla_w J(w_t) \right\|_2^2$$

$$+ 4 \left\| (F_t(w_t) + \lambda I)^{-1} - (F(w_t) + \lambda I)^{-1} \right\|_2^2 \left\| \nabla_w J(w_t) \right\|_2^2$$

$$\leq \frac{18}{\lambda^2} \left\| v_t(\theta_t) - \nabla_w J(w_t) \right\|_2^2 + 4 \left\| (F_t(w_t) + \lambda I)^{-1} - (F(w_t) + \lambda I)^{-1} \right\|_2^2 \left\| \nabla_w J(w_t) \right\|_2^2$$

$$= \frac{18}{\lambda^2} \left\| v_t(\theta_t) - \nabla_w J(w_t) \right\|_2^2 + 4 \left\| (F_t(w_t) + \lambda I)^{-1}(F(w_t) - F_t(w_t))(F(w_t) + \lambda I)^{-1} \right\|_2^2 \left\| \nabla_w J(w_t) \right\|_2^2$$

$$\leq \frac{18}{\lambda^2} \left\| v_t(\theta_t) - \nabla_w J(w_t) \right\|_2^2 + 4 \left\| (F_t(w_t) + \lambda I)^{-1} \right\|_2^2 \left\| F(w_t) - F_t(w_t) \right\|_2^2 \left\| (F(w_t) + \lambda I)^{-1} \right\|_2^2 \left\| \nabla_w J(w_t) \right\|_2^2$$

$$\leq \frac{18}{\lambda^2} \left\| v_t(\theta_t) - \nabla_w J(w_t) \right\|_2^2 + \frac{4 r_{\max}^2}{\lambda^4 (1 - \gamma)^2} \left\| F(w_t) - F_t(w_t) \right\|_2^2. \tag{33}$$

Substituting eq. (33) into eq. (32), rearranging the terms and taking expectation on both sides conditioned over $\mathcal{F}_t$ yield

$$\left( \frac{\alpha}{2(1 + \lambda)} - \frac{L_J \alpha^2}{\lambda^2} \right) \mathbb{E}[\|\nabla_w J(w_t)\|_2^2 | \mathcal{F}_t]$$

$$\leq \mathbb{E}[J(w_{t+1})|\mathcal{F}_t] - J(w_t) + \left(\frac{\alpha(1+\lambda)}{2} + L_J\alpha^2\right)\frac{18}{\lambda^2}\mathbb{E}[\|v_t(\theta_t) - \nabla_w J(w_t)\|_2^2 \,|\mathcal{F}_t]$$

$$+ \left(\frac{\alpha(1+\lambda)}{2} + L_J\alpha^2\right)\frac{4r_{\max}^2}{\lambda^4(1-\gamma)^2}\mathbb{E}[\|F(w_t) - F_t(w_t)\|_2^2 \,|\mathcal{F}_t]$$

$$\overset{(i)}{\leq} \mathbb{E}[J(w_{t+1})|\mathcal{F}_t] - J(w_t) + \left(\frac{\alpha(1+\lambda)}{2} + L_J\alpha^2\right)\frac{4r_{\max}^2}{\lambda^4(1-\gamma)^2}\frac{8[1+(\kappa-1)\rho]}{(1-\rho)B}$$

$$+ \frac{18}{\lambda^2}\left(\frac{\alpha(1+\lambda)}{2} + L_J\alpha^2\right)\left(\frac{24(r_{\max}+2R_\theta)^2[1+(\kappa-1)\rho]}{(1-\rho)B} + 6\left\|\theta_t - \theta_{w_t}^*\right\|_2^2 + 12\zeta_{\text{approx}}^{\text{critic}}\right),$$

where $(i)$ follows from eq. (27) and the fact that

$$\mathbb{E}[\|F(w_t) - F_t(w_t)\|_2^2 \,|\mathcal{F}_t] \leq \frac{8[1+(\kappa-1)\rho]}{(1-\rho)B} \quad \text{(implied by Lemma 2).} \tag{34}$$

Letting $\alpha = \frac{\lambda^2}{4L_J(1+\lambda)}$, we obtain

$$\frac{\alpha}{4(1+\lambda)}\mathbb{E}[\|\nabla_w J(w_t)\|_2^2 \,|\mathcal{F}_t]$$

$$\leq \mathbb{E}[J(w_{t+1})|\mathcal{F}_t] - J(w_t) + \left(\frac{\alpha(1+\lambda)}{2} + L_J\alpha^2\right)\left(\frac{32r_{\max}^2}{\lambda^4(1-\gamma)^2} + \frac{432(r_{\max}+2R_\theta)^2}{\lambda^2}\right)\frac{1+(\kappa-1)\rho}{(1-\rho)B}$$

$$+ \frac{108}{\lambda^2}\left(\frac{\alpha(1+\lambda)}{2} + L_J\alpha^2\right)\left\|\theta_t - \theta_{w_t}^*\right\|_2^2 + \frac{216}{\lambda^2}\left(\frac{\alpha(1+\lambda)}{2} + L_J\alpha^2\right)\zeta_{\text{approx}}^{\text{critic}}. \tag{35}$$

Taking expectation over $\mathcal{F}_t$ on both sides of eq. (35) and then taking the summation over $t = \{0, \cdots, T-1\}$ yield

$$\frac{\alpha}{4(1+\lambda)}\sum_{t=0}^{T-1}\mathbb{E}[\|\nabla_w J(w_t)\|_2^2]$$

$$\leq \mathbb{E}[J(w_T)] - J(w_0) + T\left(\frac{\alpha(1+\lambda)}{2} + L_J\alpha^2\right)\left(\frac{32r_{\max}^2}{\lambda^4(1-\gamma)^2} + \frac{432(r_{\max}+2R_\theta)^2}{\lambda^2}\right)\frac{1+(\kappa-1)\rho}{(1-\rho)B}$$

$$+ \frac{108}{\lambda^2}\left(\frac{\alpha(1+\lambda)}{2} + L_J\alpha^2\right)\sum_{t=0}^{T-1}\mathbb{E}\left[\left\|\theta_t - \theta_{w_t}^*\right\|_2^2\right] + \frac{216T}{\lambda^2}\left(\frac{\alpha(1+\lambda)}{2} + L_J\alpha^2\right)\zeta_{\text{approx}}^{\text{critic}}.$$
$$\tag{36}$$

Dividing both sides of eq. (36) by $\frac{\alpha T}{4(1+\lambda)}$ yields

$$\frac{1}{T}\sum_{t=0}^{T-1}\mathbb{E}[\|\nabla_w J(w_t)\|_2^2]$$

$$\leq \frac{16L_J(1+\lambda)^2}{\lambda^2}\frac{\mathbb{E}[J(w_T)] - J(w_0)}{T} + \frac{108}{\lambda^2}\left[2(1+\lambda)^2 + \lambda^2\right]\frac{\sum_{t=0}^{T-1}\mathbb{E}\left[\left\|\theta_t - \theta_{w_t}^*\right\|_2^2\right]}{T}$$

$$+ \left[2(1+\lambda)^2 + \lambda^2\right]\left(\frac{32r_{\max}^2}{\lambda^4(1-\gamma)^2} + \frac{432(r_{\max}+2R_\theta)^2}{\lambda^2}\right)\frac{1+(\kappa-1)\rho}{(1-\rho)B}$$

$$+ \frac{216}{\lambda^2}\left[2(1+\lambda)^2 + \lambda^2\right]\zeta_{\text{approx}}^{\text{critic}}. \tag{37}$$

Then, given the above convergence result on the gradient norm, we proceed to prove the convergence of NAC in terms of the function value. Denote $D(w) = D_{KL}\left(\pi^*(\cdot|s), \pi_w(\cdot|s)\right) = \mathbb{E}_{\nu_{\pi^*}}\left[\log\frac{\pi^*(a|s)}{\pi_w(a|s)}\right]$, $u_{w_t}^\lambda = (F(w_t) + \lambda I)^{-1}\nabla_w J(w_t)$ and $u_{w_t}^\dagger = F(w_t)^\dagger \nabla_w J(w_t)$. We proceed as follows:

$$D(w_t) - D(w_{t+1})$$

$$= \mathbb{E}_{\nu_{\pi^*}}\left[\log(\pi_{w_{t+1}}(a|s)) - \log(\pi_{w_t}(a|s))\right]$$

$$\overset{(i)}{\geq} \mathbb{E}_{\nu_{\pi^*}} \left[ \nabla_w \log(\pi_{w_t}(a|s)) \right]^\top (w_{t+1} - w_t) - \frac{L_\psi}{2} \|w_{t+1} - w_t\|_2^2$$

$$= \mathbb{E}_{\nu_{\pi^*}} \left[ \psi_{w_t}(s,a) \right]^\top (w_{t+1} - w_t) - \frac{L_\psi}{2} \|w_{t+1} - w_t\|_2^2$$

$$= \alpha \mathbb{E}_{\nu_{\pi^*}} \left[ \psi_{w_t}(s,a) \right]^\top u_t(\theta_t) - \frac{L_\psi}{2} \alpha^2 \|u_t(\theta_t)\|_2^2$$

$$= \alpha \mathbb{E}_{\nu_{\pi^*}} \left[ \psi_{w_t}(s,a) \right]^\top u_{w_t}^\lambda + \alpha \mathbb{E}_{\nu_{\pi^*}} \left[ \psi_{w_t}(s,a) \right]^\top (u_t(\theta_t) - u_{w_t}^\lambda) - \frac{L_\psi}{2} \alpha^2 \|u_t(\theta_t)\|_2^2$$

$$= \alpha \mathbb{E}_{\nu_{\pi^*}} \left[ \psi_{w_t}(s,a) \right]^\top u_{w_t}^\dagger + \alpha \mathbb{E}_{\nu_{\pi^*}} \left[ \psi_{w_t}(s,a) \right]^\top (u_{w_t}^\lambda - u_{w_t}^\dagger) + \alpha \mathbb{E}_{\nu_{\pi^*}} \left[ \psi_{w_t}(s,a) \right]^\top (u_t(\theta_t) - u_{w_t}^\lambda)$$
$$- \frac{L_\psi}{2} \alpha^2 \|u_t(\theta_t)\|_2^2$$

$$= \alpha \mathbb{E}_{\nu_{\pi^*}} \left[ A_{\pi_{w_t}}(s,a) \right] + \alpha \mathbb{E}_{\nu_{\pi^*}} \left[ \psi_{w_t}(s,a) \right]^\top (u_{w_t}^\lambda - u_{w_t}^\dagger) + \alpha \mathbb{E}_{\nu_{\pi^*}} \left[ \psi_{w_t}(s,a) \right]^\top (u_t(\theta_t) - u_{w_t}^\lambda)$$
$$+ \alpha \mathbb{E}_{\nu_{\pi^*}} \left[ \psi_{w_t}(s,a)^\top u_{w_t}^\dagger - A_{\pi_{w_t}}(s,a) \right] - \frac{L_\psi}{2} \alpha^2 \|u_t(\theta_t)\|_2^2$$

$$\overset{(ii)}{=} (1-\gamma)\alpha \Big( J(\pi^*) - J(\pi_{w_t}) \Big) + \alpha \mathbb{E}_{\nu_{\pi^*}} \left[ \psi_{w_t}(s,a) \right]^\top (u_{w_t}^\lambda - u_{w_t}^\dagger) + \alpha \mathbb{E}_{\nu_{\pi^*}} \left[ \psi_{w_t}(s,a) \right]^\top (u_t(\theta_t) - u_{w_t}^\lambda)$$
$$+ \alpha \mathbb{E}_{\nu_{\pi^*}} \left[ \psi_{w_t}(s,a)^\top u_{w_t}^\dagger - A_{\pi_{w_t}}(s,a) \right] - \frac{L_\psi}{2} \alpha^2 \|u_t(\theta_t)\|_2^2$$

$$\geq (1-\gamma)\alpha \Big( J(\pi^*) - J(\pi_{w_t}) \Big) + \alpha \mathbb{E}_{\nu_{\pi^*}} \left[ \psi_{w_t}(s,a) \right]^\top (u_{w_t}^\lambda - u_{w_t}^\dagger) + \alpha \mathbb{E}_{\nu_{\pi^*}} \left[ \psi_{w_t}(s,a) \right]^\top (u_t(\theta_t) - u_{w_t}^\lambda)$$
$$- \alpha \sqrt{\mathbb{E}_{\nu_{\pi^*}} \left[ \psi_{w_t}(s,a)^\top u_{w_t}^\dagger - A_{\pi_{w_t}}(s,a) \right]^2} - \frac{L_\psi}{2} \alpha^2 \|u_t(\theta_t)\|_2^2$$

$$\overset{(iii)}{\geq} (1-\gamma)\alpha \Big( J(\pi^*) - J(\pi_{w_t}) \Big) + \alpha \mathbb{E}_{\nu_{\pi^*}} \left[ \psi_{w_t}(s,a) \right]^\top (u_{w_t}^\lambda - u_{w_t}^\dagger) + \alpha \mathbb{E}_{\nu_{\pi^*}} \left[ \psi_{w_t}(s,a) \right]^\top (u_t(\theta_t) - u_{w_t}^\lambda)$$
$$- \sqrt{\left\| \frac{\nu_{\pi^*}}{\nu_{\pi_{w_t}}} \right\|_\infty} \alpha \sqrt{\mathbb{E}_{\nu_{\pi_{w_t}}} \left[ \psi_{w_t}(s,a)^\top u_{w_t}^\dagger - A_{\pi_{w_t}}(s,a) \right]^2} - \frac{L_\psi}{2} \alpha^2 \|u_t(\theta_t)\|_2^2$$

$$\overset{(iv)}{\geq} (1-\gamma)\alpha \Big( J(\pi^*) - J(\pi_{w_t}) \Big) + \alpha \mathbb{E}_{\nu_{\pi^*}} \left[ \psi_{w_t}(s,a) \right]^\top (u_{w_t}^\lambda - u_{w_t}^\dagger) + \alpha \mathbb{E}_{\nu_{\pi^*}} \left[ \psi_{w_t}(s,a) \right]^\top (u_t(\theta_t) - u_{w_t}^\lambda)$$
$$- \sqrt{\frac{1}{1-\gamma} \left\| \frac{\nu_{\pi^*}}{\nu_{\pi_{w_0}}} \right\|_\infty} \alpha \sqrt{\mathbb{E}_{\nu_{\pi_{w_t}}} \left[ \psi_{w_t}(s,a)^\top u_{w_t}^\dagger - A_{\pi_{w_t}}(s,a) \right]^2} - \frac{L_\psi}{2} \alpha^2 \|u_t(\theta_t)\|_2^2$$

$$\overset{(v)}{\geq} (1-\gamma)\alpha \Big( J(\pi^*) - J(\pi_{w_t}) \Big) - \alpha C_r \lambda - \alpha \left\| u_t(\theta_t) - u_{w_t}^\lambda \right\|_2$$
$$- \alpha \sqrt{\frac{1}{1-\gamma} \left\| \frac{\nu_{\pi^*}}{\nu_{\pi_{w_0}}} \right\|_\infty} \sqrt{\mathbb{E}_{\nu_{\pi_{w_t}}} \left[ \psi_{w_t}(s,a)^\top u_{w_t}^\dagger - A_{\pi_{w_t}}(s,a) \right]^2} - \frac{L_\psi}{2} \alpha^2 \|u_t(\theta_t)\|_2^2, \qquad (38)$$

where $(i)$ follows from the $L_\psi$-Lipschitz condition indicated in Lemma 5, $(ii)$ follows because

$$\mathbb{E}_{\nu_{\pi^*}}[A_{\pi_{w_t}}(s,a)] = (1-\gamma)\Big( J(\pi^*) - J(\pi_{w_t}) \Big),$$

in Lemma 3.2 of [1], $(iii)$ follows from the fact that

$$\left\| \frac{\nu_{\pi^*}}{\nu_{\pi_{w_t}}} \right\|_\infty \mathbb{E}_{\nu_{\pi_{w_t}}} \left[ \psi_{w_t}(s,a)^\top u_{w_t}^\dagger - A_{\pi_{w_t}}(s,a) \right]^2 \geq \mathbb{E}_{\nu_{\pi^*}} \left[ \psi_{w_t}(s,a)^\top u_{w_t}^\dagger - A_{\pi_{w_t}}(s,a) \right]^2,$$

$(iv)$ follows because $\nu_{\pi_{w_t}} \geq (1-\gamma)\nu_{\pi_{w_0}}$ in [1, 17], and $(v)$ follows from Lemma 6. Recalling the definition $\zeta_{\text{approx}}^{\text{actor}} = \max_{w \in \mathcal{W}} \min_{p \in \mathbb{R}^{d_2}} \mathbb{E}_{\nu_{\pi_w}} \left[ \psi_w(s,a)^\top p - A_{\pi_w}(s,a) \right]^2$, we have

$$D(w_t) - D(w_{t+1})$$
$$\geq (1-\gamma)\alpha \Big( J(\pi^*) - J(\pi_{w_t}) \Big) - \alpha C_r \lambda - \alpha \left\| u_t(\theta_t) - u_{w_t}^\lambda \right\|_2$$

$$
- \alpha \sqrt{\frac{1}{1-\gamma} \left\| \frac{\nu_{\pi^*}}{\nu_{\pi_{w_0}}} \right\|_{\infty}} \sqrt{\zeta_{\text{approx}}^{\text{actor}}} - \frac{L_\psi}{2} \alpha^2 \left\| u_t(\theta_t) \right\|_2^2
$$

$$
\geq (1-\gamma)\alpha \Big( J(\pi^*) - J(\pi_{w_t}) \Big) - \alpha C_r \lambda - \alpha \left\| u_t(\theta_t) - u_{w_t}^\lambda \right\|_2
$$

$$
- \alpha \sqrt{\frac{1}{1-\gamma} \left\| \frac{\nu_{\pi^*}}{\nu_{\pi_{w_0}}} \right\|_{\infty}} \sqrt{\zeta_{\text{approx}}^{\text{actor}}} - L_\psi \alpha^2 \left\| u_t(\theta_t) - u_{w_t}^\lambda \right\|_2^2 - L_\psi \alpha^2 \left\| u_{w_t}^\lambda \right\|_2^2
$$

$$
\geq (1-\gamma)\alpha \Big( J(\pi^*) - J(\pi_{w_t}) \Big) - \alpha C_r \lambda - \alpha \left\| u_t(\theta_t) - u_{w_t}^\lambda \right\|_2
$$

$$
- \alpha \sqrt{\frac{1}{1-\gamma} \left\| \frac{\nu_{\pi^*}}{\nu_{\pi_{w_0}}} \right\|_{\infty}} \sqrt{\zeta_{\text{approx}}^{\text{actor}}} - L_\psi \alpha^2 \left\| u_t(\theta_t) - u_{w_t}^\lambda \right\|_2^2 - \frac{L_\psi \alpha^2}{\lambda^2} \left\| \nabla_w J(w_t) \right\|_2^2 .
$$

$$(39)$$

Rearranging eq. (39), dividing both sides by $(1-\gamma)\alpha$, and taking expectation on both sides yield

$$
J(\pi^*) - \mathbb{E}[J(\pi_{w_t})]
$$

$$
\leq \frac{\mathbb{E}[D(w_t)] - \mathbb{E}[D(w_{t+1})]}{(1-\gamma)\alpha} + \frac{\mathbb{E}[\left\| u_t(\theta_t) - u_{w_t}^\lambda \right\|_2]}{1-\gamma} + \sqrt{\frac{1}{(1-\gamma)^3} \left\| \frac{\nu_{\pi^*}}{\nu_{\pi_{w_0}}} \right\|_{\infty}} \sqrt{\zeta_{\text{approx}}^{\text{actor}}}
$$

$$
+ \frac{L_\psi \alpha \mathbb{E}[\left\| u_t(\theta_t) - u_{w_t}^\lambda \right\|_2^2]}{1-\gamma} + \frac{L_\psi \alpha}{(1-\gamma)\lambda^2} \mathbb{E}[\left\| \nabla_w J(w_t) \right\|_2^2] + \frac{C_r \lambda}{1-\gamma}
$$

$$
\leq \frac{\mathbb{E}[D(w_t)] - \mathbb{E}[D(w_{t+1})]}{(1-\gamma)\alpha} + \frac{\sqrt{\mathbb{E}[\left\| u_t(\theta_t) - u_{w_t}^\lambda \right\|_2^2]}}{1-\gamma} + \frac{L_\psi \alpha \mathbb{E}[\left\| u_t(\theta_t) - u_{w_t}^\lambda \right\|_2^2]}{1-\gamma}
$$

$$
+ \frac{L_\psi \alpha}{(1-\gamma)\lambda^2} \mathbb{E}[\left\| \nabla_w J(w_t) \right\|_2^2] + \sqrt{\frac{1}{(1-\gamma)^3} \left\| \frac{\nu_{\pi^*}}{\nu_{\pi_{w_0}}} \right\|_{\infty}} \sqrt{\zeta_{\text{approx}}^{\text{actor}}} + \frac{C_r \lambda}{1-\gamma} . \tag{40}
$$

Recalling eq. (33), we have

$$
\mathbb{E}[\left\| u_t(\theta_t) - u_{w_t}^\lambda \right\|_2^2] = \mathbb{E}[\left\| u_t(\theta_t) - (F(w_t) + \lambda I)^{-1} \nabla_w J(w_t) \right\|_2^2]
$$

$$
\leq \frac{18}{\lambda^2} \mathbb{E}[\left\| v_t(\theta_t) - \nabla_w J(w_t) \right\|_2^2] + \frac{4 r_{\max}^2}{\lambda^4 (1-\gamma)^2} \mathbb{E}[\left\| F(w_t) - F_t(w_t) \right\|_2^2]
$$

$$
\overset{(i)}{\leq} \frac{18}{\lambda^2} \left[ \frac{24(r_{\max} + 2R_\theta)^2 [1 + (\kappa-1)\rho]}{B(1-\rho)} + 6\mathbb{E}[\left\| \theta_t - \theta_{w_t}^* \right\|_2^2] + 12\zeta_{\text{approx}}^{\text{critic}} \right]
$$

$$
+ \frac{4 r_{\max}^2}{\lambda^4 (1-\gamma)^2} \frac{8[1 + (\kappa-1)\rho]}{(1-\rho)B} ,
$$

where $(i)$ follows from eq. (27) and eq. (34). Letting $\mathbb{E}[\left\| \theta_t - \theta_{w_t}^* \right\|_2^2] \leq \frac{\zeta_{\text{approx}}^{\text{critic}}}{64}$ and

$$
B \geq \max \left\{ \frac{24(r_{\max} + 2R_\theta)^2 [1 + (\kappa-1)\rho]}{(1-\rho)\zeta_{\text{approx}}^{\text{critic}}}, \frac{8 r_{\max}^2 [1 + (\kappa-1)\rho]}{\lambda^2 (1-\gamma)^2 (1-\rho)\zeta_{\text{approx}}^{\text{critic}}} \right\} ,
$$

we have

$$
\mathbb{E}[\left\| u_t(\theta_t) - u_{w_t}^\lambda \right\|_2^2] \leq \frac{256}{\lambda^2} \zeta_{\text{approx}}^{\text{critic}} . \tag{41}
$$

Note that $\zeta_{\text{approx}}^{\text{critic}}$ is not small in general. Without loss of generality, here we assume $\zeta_{\text{approx}}^{\text{critic}} = \Theta(1)$. Substituting eq. (41) into eq. (40) yields

$$
J(\pi^*) - \mathbb{E}[J(\pi_{w_t})]
$$

$$
\leq \frac{\mathbb{E}[D(w_t)] - \mathbb{E}[D(w_{t+1})]}{(1-\gamma)\alpha} + \frac{16\sqrt{\zeta_{\text{approx}}^{\text{critic}}}}{\lambda(1-\gamma)} + \frac{256 L_\psi \alpha \zeta_{\text{approx}}^{\text{critic}}}{\lambda^2 (1-\gamma)} + \frac{L_\psi \alpha}{(1-\gamma)\lambda^2} \mathbb{E}[\left\| \nabla_w J(w_t) \right\|_2^2]
$$

$$+ \sqrt{\frac{1}{(1-\gamma)^3}\left\|\frac{\nu_{\pi^*}}{\nu_{\pi_{w_0}}}\right\|_\infty}\sqrt{\zeta_{\text{approx}}^{\text{actor}}} + \frac{C_r\lambda}{1-\gamma}. \tag{42}$$

Substituting the value of $\alpha$ into eq. (42), taking summation of eq. (42) over $t = \{0,\cdots,T-1\}$, and dividing both sides by $T$ yield

$$J(\pi^*) - \frac{1}{T}\sum_{t=0}^{T-1}\mathbb{E}[J(\pi_{w_t})]$$

$$\leq \frac{4L_J(1+\lambda)(D(w_0) - \mathbb{E}[D(w_T)])}{T(1-\gamma)\lambda^2} + \frac{L_\psi}{4(1-\gamma)L_J(C_\psi^2+\lambda)}\frac{\sum_{t=0}^{T-1}\mathbb{E}[\|\nabla_w J(w_t)\|_2^2]}{T}$$

$$+ \frac{16\sqrt{\zeta_{\text{approx}}^{\text{critic}}}}{\lambda(1-\gamma)} + \frac{64L_\psi\zeta_{\text{approx}}^{\text{critic}}}{(1-\gamma)L_J(1+\lambda)} + \sqrt{\frac{1}{(1-\gamma)^3}\left\|\frac{\nu_{\pi^*}}{\nu_{\pi_{w_0}}}\right\|_\infty}\sqrt{\zeta_{\text{approx}}^{\text{actor}}} + \frac{C_r\lambda}{1-\gamma}$$

$$\overset{(i)}{\leq} \frac{4L_J(1+\lambda)(D(w_0) - \mathbb{E}[D(w_T)])}{T(1-\gamma)\lambda^2} + \frac{4L_\psi(1+\lambda)}{\lambda^2(1-\gamma)}\frac{\mathbb{E}[J(w_T)] - J(w_0)}{T}$$

$$+ \frac{81L_\psi(1+\lambda)}{\lambda^2(1-\gamma)L_J}\frac{\sum_{t=0}^{T-1}\mathbb{E}\left[\left\|\theta_t - \theta_{w_t}^*\right\|_2^2\right]}{T}$$

$$+ \frac{3L_\psi(1+\lambda)}{(1-\gamma)L_J}\left(\frac{8r_{\max}^2}{\lambda^4(1-\gamma)^2} + \frac{108(r_{\max}+2R_\theta)^2}{\lambda^2}\right)\frac{1+(\kappa-1)\rho}{(1-\rho)B}$$

$$+ \frac{162L_\psi(1+\lambda)}{\lambda^2(1-\gamma)L_J}\zeta_{\text{approx}}^{\text{critic}} + \frac{16\sqrt{\zeta_{\text{approx}}^{\text{critic}}}}{\lambda(1-\gamma)} + \frac{64L_\psi\zeta_{\text{approx}}^{\text{critic}}}{(1-\gamma)L_J(1+\lambda)}$$

$$+ \sqrt{\frac{1}{(1-\gamma)^3}\left\|\frac{\nu_{\pi^*}}{\nu_{\pi_{w_0}}}\right\|_\infty}\sqrt{\zeta_{\text{approx}}^{\text{actor}}} + \frac{C_r\lambda}{1-\gamma},$$

where $(i)$ follows from eq. (37). Furthermore, letting

$$T \geq \max\left\{\frac{16L_J(1+\lambda)}{\epsilon(1-\gamma)\lambda^2}, \frac{16r_{\max}L_\psi(1+\lambda)}{\epsilon(1-\gamma)^2\lambda^2}\right\},$$

$$B \geq \frac{3L_\psi(1+\lambda)}{\epsilon(1-\gamma)L_J}\left(\frac{32r_{\max}^2}{\lambda^4(1-\gamma)^2} + \frac{432(r_{\max}+2R_\theta)^2}{\lambda^2}\right)\frac{1+(\kappa-1)\rho}{(1-\rho)},$$

$$\mathbb{E}\left[\left\|\theta_t - \theta_{w_t}^*\right\|_2^2\right] \leq \frac{\epsilon\lambda^2(1-\gamma)L_J}{324L_\psi(1+\lambda)}, \quad \text{for all} \quad 0 \geq t \geq T-1,$$

$$\lambda = \sqrt{\zeta_{\text{approx}}^{\text{critic}}},$$

we have

$$J(\pi^*) - \frac{1}{T}\sum_{t=0}^{T-1}\mathbb{E}[J(\pi_{w_t})] \leq \epsilon + \mathcal{O}\left(\frac{\sqrt{\zeta_{\text{approx}}^{\text{actor}}}}{(1-\gamma)^{1.5}}\right) + \mathcal{O}\left(\frac{\sqrt{\zeta_{\text{approx}}^{\text{critic}}}}{1-\gamma}\right).$$

The total sample complexity is given by

$$(B + MT_c)T = \mathcal{O}\left[\left(\frac{1}{(1-\gamma)^2\epsilon} + \frac{1}{\epsilon}\log\left(\frac{1}{\epsilon}\right)\right)\frac{1}{(1-\gamma)^2\epsilon}\right]$$

$$= \mathcal{O}\left(\frac{1}{(1-\gamma)^4\epsilon^2}\log\left(\frac{1}{\epsilon}\right)\right).$$

$\square$