[Reviews · NeurIPS 2020]

Review 1

Summary and Contributions: This paper analyzes convergence rates and sample complexity for actor-critic algorithms under linear function approximation for the critic under batched updates for the critic and the actor in an infinite-horizon discounted setting. Their main technical idea is to use a constant step size while controlling for batch bias error and hence obtain better sample complexity over single-sample methods.

Strengths: The paper introduces some interesting techniques which could be of interest besides the scope of this work. I looked over some proofs in the appendix and the theoretical claims seem correct, and interesting overall. The significance is that they could show that AC algorithms perform better than PG algorithms. In this sense, the contribution is significant and interesting.

Weaknesses: I think that while the authors compared to existing analyses of PG based methods and showed that their analyses of AC and NAC are better than their PG counterparts via a new style of analysis, I wonder if their analysis tricks when applied to PG methods improve their guarantees too? And if their analysis tricks does improve PG guarantees too, how does it compare then? It would be good if authors could elaborate on this aspect more. Is this a question of 1/B vs 1/\sqrt{B}?

Correctness: The theorem proofs seem correct to me (but that said, I haven't verified each step of the proof).

Clarity: The paper is well written, although I would suggest adding some more definitions, like what is meant by "single sample SA", or what is meant by "markovian sampling" -- these are obvious, but it is good to state that once.

Relation to Prior Work: I think the discussion of prior work is decent, and the differences are brought out decently well.

Reproducibility: Yes

Additional Feedback: Main comments are in the other questions. I would like a description of whether their trick improves guarantees for policy gradient methods too, agnostic of the critic under their chosen assumptions. ----------------------- Thanks for the response. Your answer for why these techniques may not be applicable to policy gradient methods makes sense, and it would be great to add this to the final paper in my opinion -- to distinguish the potential confounding factor of better analysis for AC/NAC giving better bounds when compared to not so great analysis of PG. A toy example pointed out by R3 should be included as well, and some quantification of the critic approximation error (R4) would be good to discuss as well.


Review 2

Summary and Contributions: This paper is on improving sample complexity bounds for actor-critic (AC) and natural AC (NAC). The paper gives the first sample complexity bounds for actor-critic methods under Markovian sampling (non-iid) and first results showing AC and NAC have tighter sample complexity bounds than policy gradient and natural policy gradient algorithms (the latter better reflects empirical observations). The three core theoretical results of the paper are: 1. Sample complexity bound for mini-batch TD 2. Sample complexity bound for mini-batch AC 3. Sample complexity bound for mini-batch NAC All results are derived under Markovian sampling as opposed to iid sampling used in other methods.

Strengths: Overall, I like the paper and like to see results moving theoretical bounds closer to practical settings for these algorithms. I believe the contributions are significant, novel, and relevant to the theoretical RL community.

Weaknesses: Admittedly this is a bit nitpicky, but it would be interesting to complement the theoretical results with empirical results in toy problems (maybe random MDPs?). Such results can give an idea of how tight novel bounds are in practice and could also be a way to give better insight into how parameters like mini-batch size affect bounds. With that said, I don't see this as a major limitation of the work.

Correctness: The paper's arguments are presented entirely theoretically. As far as I can tell the proofs of results are correct.

Clarity: The paper is clearly written. I have some minor comments and questions given as feedback to the authors.

Relation to Prior Work: Yes, the authors include a related work section and a table showing how their contributions relate to existing work.

Reproducibility: Yes

Additional Feedback: Thank you for your response. I've left my review as is. Questions for authors: The section on mini-batch TD learning discusses bias error. I'm not sure bias error is the right way to describe this, particularly w.r.t. the part that disappears with a larger mini-batch (line 211). Isn't this more of a variance error? Does Thm 1 depend on both assumptions or just assumption 2? In assumption 1, should L_phi be L_psi? Is there a typo in Assumption 2's equation? What is P(s_t \in \cdot | s_0=s)? Minor comments: 37: infinity -> infinite Don't use numerical citations as nouns. Better to give author names.


Review 3

Summary and Contributions: The paper provides finite-time analysis for actor critic (AC) and natural actor critic algorithms (NAC) under the conditions of infinite horizon, Markovian sampling, mini-batch training, general actor approximation, and linear critic approximation. The presented convergence rates outperform existing results (that do not consider mini-batch training).

Strengths: 1. The paper is the first to analyze Markovian min-batch training for AC and NAC. Mini-batch training is rarely studied but is widely used in practice, so the contribution is both novel and significant for theoretical analysis. 2. The presented techniques (linear and nonlinear stochastic analysis) are demonstrated to improve upon prior convergence rates, and thus provide new insights into theoretical analysis.

Weaknesses: Though the paper demonstrates faster convergence of AC compared to policy gradient (PG), its proof assumes a linear critic, which can introduce a large approximation error in practice (see Theorem 2). It is unclear whether the gain in convergence speed outweighs the approximation error.

Correctness: I have read through the main text, but only skimmed the appendix due to lack of time and domain expertise. The overall proof appears correct, but it is possible that I have missed important details.

Clarity: The paper is very well written. The presentation is easy to follow. Main ideas are summarized and explained concisely.

Relation to Prior Work: The paper clearly shows differences from previous contributions in terms of mini-batch training and Markovian sampling.

Reproducibility: Yes

Additional Feedback: The authors’ response has addressed my questions. I will keep my score. ---------------------------- 1. Can nonlinear SA be applied to a nonlinear critic as well? This is a natural question to ask, so it could be worth an explanation somewhere. 2. In practice, some have noticed improved convergence rates of NAC compared to AC (e.g. ACKTR v.s. A2C in [1]). However, this paper suggests a slower rate by a factor of (1-\gamma)^{-2}. What could cause the difference and how could the theory here guide development of deep RL algorithms? [1] Wu, Y., Mansimov, E., Grosse, R.B., Liao, S. and Ba, J., 2017. Scalable trust-region method for deep reinforcement learning using kronecker-factored approximation. In Advances in neural information processing systems (pp. 5279-5288).

[Author Response · NeurIPS 2020]

**Reviewer 1: Q1:** I wonder if their analysis tricks of AC/NAC when applied to PG methods improve their guarantees too? If their analysis tricks do improve PG guarantees, how does it compare then? Is this a question of $1/B$ vs $1/\sqrt{B}$?

**A1:** Great question! This paper proposed two major tricks to improve the convergence rate of AC/NAC: **Trick I** of analysis of mini-batch sampling and **Trick II** of exploitation of self-reduced variance.

For **PG**, the variance error is not self-reduced, and hence trick II cannot improve its convergence rate. We next check that trick I does not improve its convergence rate either. Recall the best known convergence rate of PG is given in (Xiong et al. 2020) as $\mathcal{O}\big(\frac{1}{(1-\gamma)^2\sqrt{T}}\big)$. Thus, we require $T \geq \mathcal{O}\big(\frac{1}{(1-\gamma)^4\epsilon^2}\big)$ to achieve an $\epsilon$-accurate stationary point. Note that PG algorithm further requires a Monte Carlo rollout with average length $L = \mathcal{O}\big(\frac{1}{1-\gamma}\big)$ to estimate Q-function for each sample. Thus, the sample complexity of PG is given by $TL = \mathcal{O}\big(\frac{1}{(1-\gamma)^5\epsilon^2}\big)$ as given in (Xiong et al. 2020). Now, applying trick I (minibatch sampling) to PG, we obtain the convergence rate of $\mathcal{O}\big(\frac{1}{(1-\gamma)^2T}\big) + \mathcal{O}\big(\frac{1}{(1-\gamma)^2B}\big)$. Thus, we require $T \geq \mathcal{O}\big(\frac{1}{(1-\gamma)^2\epsilon}\big)$ and $B \geq \mathcal{O}\big(\frac{1}{(1-\gamma)^2\epsilon}\big)$ to achieve an $\epsilon$-accurate stationary point. Thus, the sample complexity of minibatch PG is $TBL = \mathcal{O}\big(\frac{1}{(1-\gamma)^5\epsilon^2}\big)$, which is the same as that of PG.

For **NPG**, it can also be checked that trick I does not improve its rate. Since NPG has self-reduced variance, trick II does improve the sample complexity $\mathcal{O}\big(\frac{1}{(1-\gamma)^8\epsilon^4}\big)$ of NPG given in (Agarwal et al. 2019) to $\mathcal{O}\big(\frac{1}{(1-\gamma)^7\epsilon^3}\big)$. This improved rate of NPG is still worse than the sample complexity $\mathcal{O}\big(\frac{1}{(1-\gamma)^4\epsilon^2}\big)$ of NAC given in our paper .

**Reviewer 2: Q1:** It would be interesting to complement the theoretical results with empirical results in toy problem.

**A1:** Thanks for the suggestion! We are working on experiments and will add these results to the revision.

**Q2:** For the error term that disappears with a larger mini-batch (line 211). Isn't this more of a variance error?

**A2:** Yes, this error term should be called as variance error. we will fix it in the revision.

**Q3:** Does Thm 1 depend on both assumptions or just assumption 2?

**A3:** Thm 1 is based on (a) Assumption 2 and (b) $\|\phi(s,a)\|_2 \leq 1$ for all $(s,a)$ and $(\theta - \theta_\pi^*)^\top A_\pi(\theta - \theta_\pi^*) \leq -\lambda_A \|\theta - \theta_\pi^*\|_2^2$. Item (b) is stated in the paragraph before Thm 1, which has been justified in many previous studies.

**Q4:** In Assumption 1, should $L_\phi$ be $L_\psi$? **A4:** Yes, $L_\phi$ should be $L_\psi$.

**Q5:** What is $\mathbb{P}(s_t \in \cdot | s_0 = s)$ in Assumption 2? **A5:** $\mathbb{P}(s_t \in \cdot | s_0 = s)$ denotes the probability distribution of $s_t$ conditioned on the initial state $s_0$. The notation is confusing and we will change it. Thanks for pointing it out!

**Reviewer 3: Q1:** The proof assumes a linear critic, which can introduce an approximation error in practice (see Theorem 2). It is unclear whether the gain in convergence speed outweighs the approximation error.

**A1:** Great point! Though linear critic can introduce an approximation error, a line of theoretical studies (including this work) naturally start from linear critic because it is analytically trackable. In fact, we find our analysis here can be extended to the nonlinear critic case (see our answer to Q2 below).

**Q2:** Can nonlinear SA be applied to a nonlinear critic as well?

**A2:** Great question! Yes. For a nonlinear critic, we can utilize the algorithm of nonlinear temporal difference learning with gradient correction (nonlinear TDC) to update critic's parameter, which can be analyzed by adapting our current analysis for nonlinear SA and existing technique for linear TDC. We can then incorporate the convergence analysis for the nonlinear TDC into our current analysis framework for AC/NAC to obtain the overall convergence analysis.

**Q3:** In practice, some have noticed improved convergence rates of NAC compared to AC (e.g. ACKTR v.s. A2C in [1]). However, this paper suggests a slower rate by a factor of $(1 - \gamma)^{-2}$. (a) What could cause the difference and (b) how could the theory here guide development of deep RL algorithms?

**A3: (a)** Due to the different nature of AC and NAC, existing literature (including this paper) characterize their convergence rates by different metrics: AC by the gradient norm (as in Thm2), but NAC by the function value (as in Thm 3). Thus, the theoretical convergence rates of AC (Thm 2) and NAC (Thm 3) are not directly comparable. **(b)** Our theory here provide the following insights. First, our theory shows that NAC converges to a global optimal policy, while AC converges only to a first-order stationary point, which is likely a local optimum. This theoretical result explains practical observations that ACKTR achieves larger accumulated reward than AC, and in principle captures the advantage of NAC. Second, our theory also shows that mini-batch AC/NAC converges faster than single-sample AC/NAC, which suggests that practical implementation of AC/NAC can adopt minibatch and constant stepsize to achieve fast rate.

[Meta-Review · NeurIPS 2020]

The paper provides a meaningful theoretical contribution for Natural AC algorithms. The reviewers and myself like and agree that this paper should be accepted. I thank both the authors and reviewers for their efforts.